# PlanBench: An Extensible Benchmark for Evaluating Large Language Models on Planning and Reasoning about Change

**Karthik Valmeekam**
School of Computing & AI
Arizona State University, Tempe.
`kvalmeek@asu.edu`

**Matthew Marquez**
School of Computing & AI
Arizona State University, Tempe.
`mmarqu22@asu.edu`

**Alberto Olmo**
School of Computing & AI
Arizona State University, Tempe.
`aolmoher@asu.edu`

**Sarath Sreedharan**[*]
Department of Computer Science,
Colorado State University, Fort Collins.
`sarath.sreedharan@colostate.edu`

**Subbarao Kambhampati**
School of Computing & AI
Arizona State University, Tempe.
`rao@asu.edu`

## Abstract

Generating plans of action, and reasoning about change have long been considered a core competence of intelligent agents. It is thus no surprise that evaluating the planning and reasoning capabilities of large language models (LLMs) has become a hot topic of research. Most claims about LLM planning capabilities are however based on common sense tasks–where it becomes hard to tell whether LLMs are planning or merely retrieving from their vast world knowledge. There is a strong need for systematic and extensible planning benchmarks with sufficient diversity to evaluate whether LLMs have innate planning capabilities. Motivated by this, we propose PlanBench, an extensible benchmark suite based on the kinds of domains used in the automated planning community, especially in the International Planning Competition, to test the capabilities of LLMs in planning or reasoning about actions and change. PlanBench provides sufficient diversity in both the task domains and the specific planning capabilities. Our studies also show that on many critical capabilities–including plan generation–LLM performance falls quite short, even with the SOTA models. PlanBench can thus function as a useful marker of progress of LLMs in planning and reasoning.

## 1   Introduction

The advent of large pre-trained language models have revolutionized the field of natural language processing and have also received widespread public attention. These types of transformer-based large language models (LLMs) currently provide state-of-the-art performance in many of the standard NLP tasks. LLMs essentially predict the next word in a sentence, given a certain context and these models were originally developed to perform word sequence completion tasks. In the recent times, there

---

[*]Author was at Arizona State University during part of this work

37th Conference on Neural Information Processing Systems (NeurIPS 2023) Track on Datasets and Benchmarks.

have been anecdotal evidence and claims that they possess other capabilities that are not normally associated with sequence completion. This led to a sudden outburst of research probing and studying their behavior almost as if they were artificial organisms (c.f. [15]). In this paper, we are particularly interested in the line of research efforts that investigate (and showcase) the reasoning capabilities of Large Language models–including commonsense reasoning [31, 27, 5], logical reasoning [29], and even ethical reasoning [14]. These works have largely been suggesting that LLM's are indeed capable of doing such kinds of reasoning [17, 34, 2].

Planning is a reasoning task that has been well studied in the AI community. In its most basic form, planning involves coming up with a course of actions (policy) which when executed would take an agent from a certain initial state to a desired world state. Planning has generally been studied primarily as an inference problem on world and reward models. These models could either be specified by humans or learned by the agent by interacting with its world. In this paper, we want to look at the ability of large language models to do reasoning about actions and change involving common-sense planning tasks. We propose PlanBench, an extensible benchmark suite based on the kinds of domains used in the automated planning community, specially in International Planning Competitions (IPC) [13] to test this. Our focus on planning is spurred by not only the fact that it is a core aspect of human intelligence [26], but also that it is required for tasks considered as potential applications of LLMs including automatic code generation.

The contribution of our paper is a curriculum for evaluating planning, wherein we identify a set of related but distinct tasks, that are central for an agent to successfully perform planning and reasoning about actions and introduce a framework for developing code which is meant to auto-generate the possible queries for each. We initialize PlanBench with two IPC domains: Blocksworld and Logistics. We also provide obfuscated versions of these domains with the obfuscations being either misleading words or random alphanumeric strings. Overall, we provide ∼26250 prompts as part of our dataset. We are also actively adding other IPC domains and tasks into the benchmark. The updated version of the entire benchmark, including the tools, datasets, and the scripts to reproduce the prompts and results in this paper, can be found at `https://github.com/karthikv792/LLMs-Planning`.

We are not the first to point out the need to perform such analyses of the reasoning capabilities of GPT-3 like LLMs. For example, [19] performed an analysis of GPT-3's reasoning capabilities on some example tasks, including different commonsense reasoning tasks varying from biological reasoning to arithmetic reasoning. However, the goal of this paper is fundamentally distinct from these earlier works in multiple ways. Firstly, we are not merely trying to point out a few example cases where LLMs fail but rather help establish an assessment framework for evaluating these systems' capabilities to perform planning. While this paper reports the results of testing Instruct-GPT3 [22] and GPT-4 [21], one could use this framework to analyse other LLMs that may be fine-tuned for such tasks. Secondly, through this framework, we are also trying to eliminate the subjective aspect of analysis that forms the core part of many of these earlier efforts. Instead, we automate and perform the analyses in a mechanistic way by leveraging automated planning models and tools to generate the queries and validate the system's answers.

## 2 Related Work

To the best of our knowledge, we are the first to introduce a benchmark specifically designed to evaluate the emerging planning abilities of LLMs, if any. The initial version of our benchmark was made public almost a year back and is currently in use by various researchers.[2] We have significantly increased the scope of our benchmark (with additional test cases and domains) from the initial version. But the idea of developing benchmarks to evaluate emergent properties of LLMs is itself not new. Some prominent existing reasoning benchmarks include, BIG-BENCH [29], GSM8K [3], AQUA [18], SVAMP [23], CommonsenseQA [31] and StrategyQA [5]. However, these tasks are simple involve shallow reasoning and do not give insight into their planning capabilities. As LLMs have been able to perform well on such tasks, there has been a lot more triumphalism about their planning capabilities, which is currently being echoed in the community.

There have been significant developments in the intersection of LLMs and planning, with LLMs undertaking various roles [16]. These roles range from generating plans [11, 33] and heuristics [33, 1]

---

[2]The number of stars (103) and forks (14) on our GitHub repository serves as an indirect estimate of the level of interest in our benchmark.

to extracting planning knowledge [7]. For example, in Say-Can [1], LLMs have been used as scoring models, which can be seen as providing planning heuristics, for the actions that the embodied robot can execute. Additionally, LLMs are also being made to use feedback from users or the environment to improve their planning performance [12, 35, 25].

Our work primarily establishes an assessment framework for evaluating a wide variety of planning capabilities of LLMs. PlanBench consists of multiple tasks, each designed to evaluate a certain aspect of reasoning about actions and change. The prompts for our tasks are showcased as few-shot examples where we provide an instance and an example completion and then ask for a completion on a new instance. Further, we use the domain description to explicitly constrain the possible actions. In many everyday scenarios, we are often asked to take into consideration unforeseen limitations and constraints. Our explicit domain description allows us to introduce such challenges and forces the LLMs to go beyond merely repeating possible information about the domain they may have come across in the training data. The ability to be conditional on the prompt is critical for the general systems to be customized for the specific domain of interest.

## 3 Background

As we are interested in investigating the basic planning problem, we want look at the most fundamental planning formalism, namely the goal-directed deterministic planning problem. These kinds of problems are colloquially referred to as *classical planning problems*.

Classical Planning Problems can be mathematically represented by using the tuple $\mathcal{P} = \langle \mathcal{D}, \mathcal{I}, \mathcal{G} \rangle$. $\mathcal{D}$ is referred to as the problem domain, $I$ is the initial state and $G$ is the goal specification. The possible truth assignment over the predicates defines the state space for the planning problem. The domain is again defined by the tuple $\mathcal{D} = \langle \mathcal{F}, \mathcal{O} \rangle$. $\mathcal{F}$ corresponds to the set of fluents, i.e., the state variable used to define the state space and each fluent corresponds to a predicate with some arity, and $\mathcal{A}$ correspond to the set of actions that can be performed as part of the planning problem. Each action $a_i[\mathcal{V}] \in \mathcal{A}$ (where $a_i$ is the operator label and $\mathcal{V}$ is the variable used by the operator and each variable could be mapped to an object), can be further defined by two components, the precondition $prec[\mathcal{V}]$ which describes *when* an action can be executed and the effects $eff[\mathcal{V}]$ which defines *what happens* when an action is executed. We will assume that $prec[\mathcal{V}]$ consists of a set of predicates defined over the variables $\mathcal{V}$. An action is assumed to be executable only if its preconditions are met, i.e., the predicates in the precondition hold in the given state. The effects $eff[\mathcal{V}]$ is further defined by the tuple $\langle add[\mathcal{V}], del[\mathcal{V}] \rangle$, where $add[\mathcal{V}]$ or add effects is the set of predicates that will be set true by the action and $del[\mathcal{V}]$ or delete effects is the set of predicates that will be set false by the action. An action is said to be grounded if we replace each of the variables with an object, else it is referred to as a lifted domain model (we use a similar convention to differentiate between lifted and grounded predicates). Below is a snippet of an action from a popular benchmark problem called Blocksworld, in PDDL. The action corresponds to putting down a block on a table in that domain.

```
(:action put-down
  :parameters  (?ob)
  :precondition (holding ?ob)
  :effect (and (clear ?ob) (arm-empty) (on-table ?ob)
              (not (holding ?ob))))
```

In the above snippet, the parameter line provides the possible variables, in this case *?ob*, which can stand for possible blocks. The precondition says that you can put down a block only if you are holding it (i.e. predicate *(holding ?ob)* is true for the block). The effects tell you that after you execute the action put-down, the block will be on the table and will be clear. You won't be holding the block and the arm will be considered *empty*. The actions may additionally be associated with cost, in these cases, one could also talk about optimal plans, i.e., a plan $\pi$ is called an optimal one if no plan exists that is less costly than $\pi$.

The above description presents one of the simpler classes of planning models and can be extended in multiple ways including allowing for object typing (including type hierarchy), more complex forms of preconditions and conditional effects, not to mention supporting richer classes of planning formalisms.

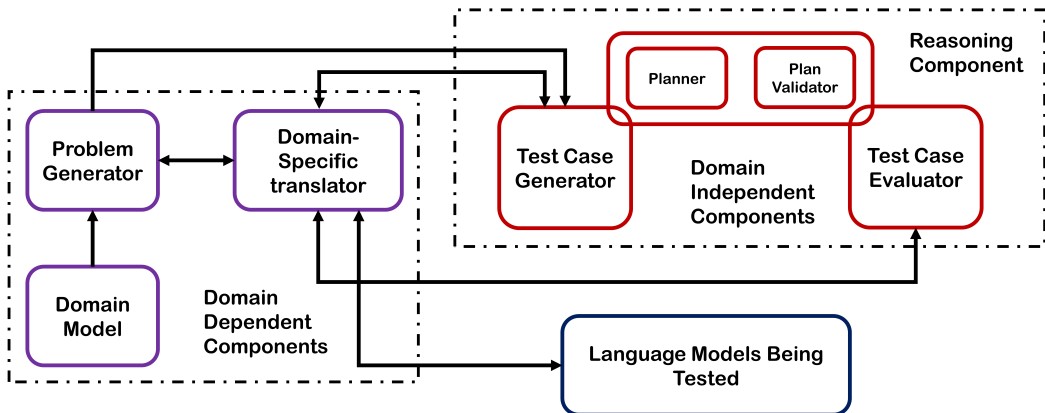

Figure 1: The diagrammatic overview of the overall test framework. Our system consists of a domain-specific component that allows the generation of various instances of the specific PDDL planning problems and the translation from PDDL to text and back. The domain-independent component is responsible for generating the test instances that will be fed into the LLM and verifying the output generated by the LLM.

## 4   Assessment Architecture

Our basic test framework consists of two categories of components, the domain-independent ones, provided as part of the framework, and the domain-dependent components which need to be developed for each new domain we test.

**Domain independent component:**   The domain-independent component is built around a planner and a plan verification component that takes various planning problems and crafts test instances corresponding to various curriculum items. This component provides the mechanism to verify the solutions generated by the LLM. The current method is going to operate almost exclusively on symbolic models (specifically ones specified using PDDL [20]) and other structured inputs compatible with such representations.

This component is responsible for generating the content for the various prompts that would be generated as part of the different test cases and for validating the output generated by the LLM. As discussed earlier, the component primarily works on formal representations of the problems, so it relies on the translator component to convert any information it generates to natural language or to convert natural language information back to formal representations. For each test case, we mainly rely on a domain-independent planner and a plan validator to generate the relevant information or to validate the output provided by the LLM. For some test cases, we incorporate domain-specific information while crafting the prompts. In each case, there is a test-case-specific component that uses the problems provided by the problem generator component to craft specific test-case content.

**Domain dependent component:**   The domain-dependent component consists of three parts; a domain model, a problem generator and a translator. The lifted domain file describes the various actions that may be available to solve any given planning problem, the various predicates that could be used to describe the various relationships over the objects that may be present at a given problem instance of the domain, and the various types of objects that may be part of the given problem. The domain model is lifted as it does not refer to the actual objects that may be part of the problem, but instead, the actions are defined independently of the exact objects it may influence.

The role of the problem generator is to generate random problem instances consisting of various objects, initial states, and goals. These problems become the basis of generating the various test cases that we will be using throughout the framework. Any distributional requirements we hope to use in the tests could be built into this problem generator.

The translator converts the symbolic model information to natural language text and *vice versa* . Translating the prompts to natural language would benefit the users of the benchmark who might want to be in the loop to either impose additional constraints or evaluate the plan themselves as these two tasks are more naturally done in natural language. For the current testbed (described below), we

developed a template-based mechanism to achieve this. In particular, we provide a natural language template for each predicate and each action, and we form texts of states and plans by concatenating these individual strings. In terms of parsing natural language text back into structured forms, the particular task we are interested in is converting plans generated by the LLM back into plan forms that can be used by plan validator tools like [10]. Since we use our prompts to shape the LLM's output, we require each action in the plan to be listed on a different line. Then, we can parse the exact action and arguments of the action by either using template-based matching or by assuming that the verb in the sentence corresponds to the action and each noun corresponds to an object which forms a parameter of the action (then mapping it to a possible action).

## 5  Current Curriculum for Testing

In this section, we go over each specific test case we provide as part of PlanBench. Each test case is meant to evaluate a central reasoning about actions and change capability and is tested in the context of a common sense planning domain. Each test case makes use of the few shot query setting of LLM where the LLM is provided a few sample answers to the specific reasoning ability being tested and is asked to respond to a new instance.[3] The exact form of the prompt will depend on the specific test cases, but every instance will start with a description of the lifted planning domain that describes what actions can be executed, their preconditions and their effects. The current set of test cases includes the following cases:

1. Plan Generation - Can the LLM come up with valid plans that will achieve a specific goal?
2. Cost Optimal Planning - Can the LLM come up with plans that are optimal to achieve a specific goal?
3. Plan Verification - Can the LLM determine if a plan will successfully execute, and if not, can it explain why?
4. Reasoning about plan execution - Can the LLM reason about what happens when a plan is executed?
5. Robustness to goal reformulation - Can the LLM recognize the same goal when specified in different ways?
6. Ability to reuse plans - Can the LLM recognize scenarios where it can reuse part or the whole of the original plan to achieve the new goal?
7. Replanning - Can the LLM replan for cases where an unexpected change is reported?
8. Plan Generalization - Can the LLM take specific plans, extract underlying procedural patterns and apply them to a new instance?

Out of the eight test cases, the first two test cases correspond to actual planning problems (i.e. plan generation and cost-optimal planning) and the rest correspond to simpler auxiliary tasks related to reasoning about action and change. We ground the test cases in multiple domains based on the kinds employed in International Planning Competitions. The domain description is included at the beginning of every prompt. In the rest of the section, we discuss the structure of the prompt for each of the test cases. We provide an example prompt and the corresponding completion generated by an LLM for each of the test cases in the Appendix.

**Plan Generation:**  Following the lifted domain description, the prompt consists of a few instances of planning problem descriptions (consisting of a description of the initial state, the goal) and the corresponding plan (which ends with a tag, henceforth referred to as the plan-end tag, that denotes the end of the plan) and finally, we end the prompt with a planning problem description. The plan-end tag (which is separate from the end-generation tag) is introduced to enable easier plan extraction in cases where the LLM also adds commentary along with the plan. In the case where our extractor cannot reasonably extract a plan from the response, we mark that instance as an incorrect one. We present an example prompt for this test case here. For the rest of the test cases, the examples are provided in the Appendix.

---

[3]While in this work we look only at one-shot natural language prompt configurations, we have also looked at various other prompt configurations (including zero-shot prompts, chain-of-thought prompts and pddl style prompts), specifically in the plan generation test case, in [33].

```
========================================
I am playing with a set of blocks where I need to arrange the blocks into stacks. Here are the
↪  actions I can do

Pick up a block
Unstack a block from on top of another block
Put down a block
Stack a block on top of another block

I have the following restrictions on my actions:
I can only pick up or unstack one block at a time.
I can only pick up or unstack a block if my hand is empty.
I can only pick up a block if the block is on the table and the block is clear. A block is clear
↪  if the block has no other blocks on top of it and if the block is not picked up.
I can only unstack a block from on top of another block if the block I am unstacking was really on
↪  top of the other block.
I can only unstack a block from on top of another block if the block I am unstacking is clear.
Once I pick up or unstack a block, I am holding the block.
I can only put down a block that I am holding.
I can only stack a block on top of another block if I am holding the block being stacked.
I can only stack a block on top of another block if the block onto which I am stacking the block
↪  is clear.
Once I put down or stack a block, my hand becomes empty.
Once you stack a block on top of a second block, the second block is no longer clear.

[STATEMENT]
As initial conditions I have that, the red block is clear, the blue block is clear, the yellow
↪  block is clear, the hand is empty, the blue block is on top of the orange block, the red block
↪  is on the table, the orange block is on the table and the yellow block is on the table.
My goal is to have that the orange block is on top of the blue block.

My plan is as follows:

[PLAN]
unstack the blue block from on top of the orange block
put down the blue block
pick up the orange block
stack the orange block on top of the blue block
[PLAN END]

[STATEMENT]
As initial conditions I have that, the red block is clear, the yellow block is clear, the hand is
↪  empty, the red block is on top of the blue block, the yellow block is on top of the orange
↪  block, the blue block is on the table and the orange block is on the table.
My goal is to have that the orange block is on top of the red block.

My plan is as follows:

[PLAN]
--------- GPT-4 response ---------
unstack the yellow block from on top of the orange block
put down the yellow block
unstack the red block from on top of the blue block
put down the red block
pick up the orange block
stack the orange block on top of the red block
[PLAN END]
--------- Extracted plan ---------
(unstack yellow orange)
(put-down yellow)
(unstack red blue)
(put-down red)
(pick-up orange)
(stack orange red)

-------- Ground truth plan ---------
(unstack yellow orange)
(put-down yellow)
(pick-up orange)
(stack orange red)
===============SUCCESS===============
```

**Cost-Optimal Planning:** The prompt is quite similar to the one used in the earlier test case with a few changes. We modify the lifted domain description by including a statement that associates a cost with each action. To make the concept of action cost better fit into common sense domains, we can

map the cost to more common concepts like the time taken for executing the action or the amount of money that needs to be spent to execute an action. In the case of each problem description, before the plan is presented we need to explicitly mention that the plan is trying to minimize cost (which depending on the scenario might correspond to saying that the plan takes the least amount of time or the plan correspond to the cheapest plan). The result generated by the LLM is evaluated similarly to the previous query, but in addition to checking if the plan is valid, we also check if the cost of the plan corresponds to the optimal plan cost.

**Plan Verification:** Plan verification involves determining whether a proposed plan will successfully execute and achieve the stated goals when applied from the given initial state. Here, the prompt will first include three example instances, the corresponding candidate plans and the verification details. The set of examples will contain one example with a goal reaching plan, one with a non-goal reaching plan and one with an inexecutable plan. The prompt then contains a new instance along with a candidate plan. The LLM is tasked with answering whether the candidate plan is valid. If it answers no, it must identify the first inexecutable action and atleast one associated missing precondition (if inexecutable) or at least one missing goal condition (if not goal reaching).

**Reasoning about plan execution:** Here the objective is not to check whether the LLM can come up with plans, or verify them, but rather if they can predict the outcome of executing an action. The prompt here again starts with the domain description, but instead of providing planning problems and plans, we provide a state, an action sequence and then the state that would result from executing that action sequence in the provided state. Finally the prompt ends with a new state and a new action sequence. The LLM is expected to come up with the resulting state, which is then checked by applying a plan executor that will try to identify what state will result from the execution of the current action sequence on the provided state. We always provide executable action sequences for this test case.

**Robustness to Goal Reformulation:** In this test case, we will see if the LLM can recognize goals it has seen before if they are slightly modified. Here the prompt remains the same as the one used for plan generation. However, all the example problems have the same initial state, and the last problem provided has not only the same initial state but also the same goal as the example problem. Here the goal may be obfuscated in a few ways, for example, the goal facts may be reordered or one might include a subset of the original goal specification (meaning the same plan would still work) or vice-versa. We can again use the same evaluation technique as the plan generation test case to validate the output.

**Ability to Reuse Plans:** In this test case, we are interested in seeing if the LLM can reuse plans or parts of plans that it has seen before. The prompt is again the same as the plan generation, but the prompt ends with a problem that can be solved by a prefix of a previously seen plan. We again keep the initial state the same across the example problems shown. The evaluation remains the same as the plan generation test case.

**Replanning:** Replanning corresponds to the problem where there may be an unexpected event that occurs while executing a plan and the system needs to come up with a new plan in response to the event. Here, we focus on the ability of the LLM to replan when unexpected changes are reported. The prompt here starts with a domain description, then a set of instances where an unexpected event occurred during execution, and a new plan in response to the event. In each instance, a planning problem and a corresponding plan are provided at the beginning, the execution of the plan is described and then an unexpected event is noted (event corresponds to some facts unexpectedly turning true or false) and then a new plan from the changed state is presented. The prompt ends with a new case where the plan after replanning is left out and the LLM is expected to complete. The evaluation involves checking whether the new plan is valid from the changed state. The LLM output is evaluated to be true if the new plan it generates achieves the goals from the unexpectedly changed state. For each of the domains, we constrain the unexpected event to be of a specific type. We describe these cases for each domain in the next section.

**Plan Generalization:** In this test case, we want to evaluate whether LLMs can recognize the underlying pattern in the plans provided in the prompt and reuse it for a new planning problem. The prompt is the same as the plan generation case, except that all plans were generated by a fixed program. Here the program may contain loops or conditional statements, but can only solve certain types of problems, that is, the initial state and goals meet certain conditions. Such programs can be

thought of as a direct generalization of line plans that we have considered in the rest of the paper [30]. Execution of this program for a specific planning problem generates a sequence of actions. In this case, we will provide some example traces generated from the program and ask LLM to come up with a plan for a new problem that could be solved by it. The evaluation again would be to take the generated plan and see if it is valid for the given problem. Since this task requires the use of specific problems that admit generalizable solutions, not every problem works for this task. As such, this task requires problems to be curated for each domain. To develop this set of problems, we select a domain-specific generalized behavior and then create several problems that can be accomplished by merely adding more iterations of the generalized behavior to an example plan. We detail our domain-specific selection of behavior in Section 6.

# 6    Dataset Details

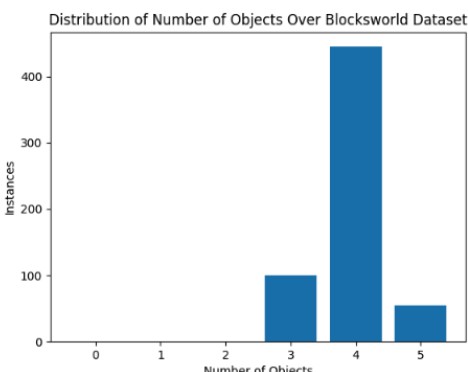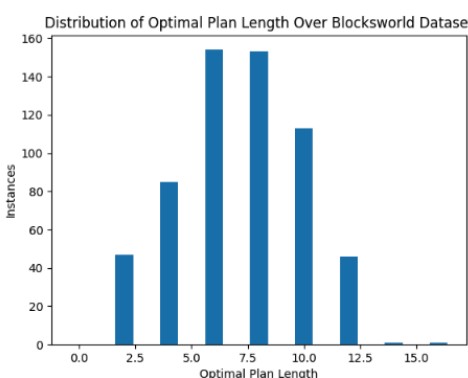

Figure 2: Distribution of the number of objects and optimal plan length for the Blocksworld problem set

While PlanBench makes it easy to run the curriculum on new domains, we initialize it with two different International Planning Competition (IPC) [13] domains: Blocksworld and Logistics. For each domain, we provide a description, distribution of selected problems, and details on domain-specific curriculum configurations below.

**Blocksworld:**  The Blocksworld domain focuses on stacking blocks on a table. One hand is available to move blocks, and only one block may be moved by the hand at a time. Blocks cannot be moved if there are blocks on top of them and blocks cannot be stacked on a block with another block already on top of it. Goals specify the order that blocks within a stack should be stacked in but may include multiple stacks or ask for blocks to be left on the table. Blocks our identified with colors. We developed 600 instances which vary in the number of objects used as well as the optimal plan length: we visualize these distributions in Figure 2. For the plan generalization test case, we focus on problems that can be solved by repeatedly adding clear blocks to the stack and generate 500 instances separate from the main blocksworld dataset. For the replanning test case we constrain the unexpected event to be of a specific type: We execute a random prefix of the plan which ensures that some block is held at the end of that prefix. We then change the resulting state by stacking the held block onto another random block which is clear and make the hand empty. This change is reported and the LLM is asked to replan from the changed state.

**Logistics:**  The Logistics domain involves moving packages between different locations. Locations are grouped by cities. Trucks can be used to move packages between locations in the same city and planes can be used to move packages between cities. Goals specify where packages should be moved. We generate 285 instances for this domain We provide the distribution of objects (which includes cities, locations, trucks, airplanes and packages) and optimal plan length over the problems in Figure 3. For the plan generalization test case we focus on problems that contain one or more instances of the following; dropping off a package at a location and picking up the next package at that same location, allowing for a chain of package movement within a city through a truck and finally have a package at an airport be flown to another airport. For the replanning test case, we execute a random prefix of the plan which ensures a package is in a truck or an airplane. We then shift the position of

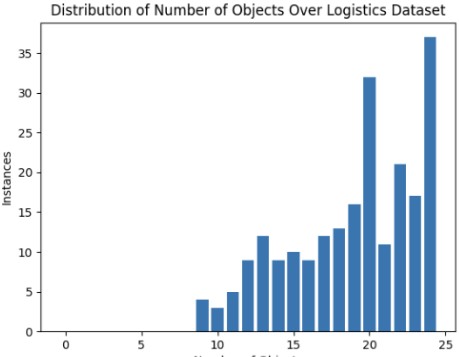
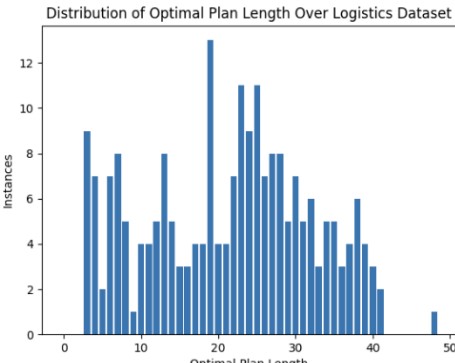

Figure 3: Distribution of the number of objects and optimal plan length for the Logistics problem set the package from inside the truck or airplane to a random location which is not the goal location. This change is reported and the LLM is asked to replan from the changed state.

**Obfuscation of domains:**  Although the domain specification is part of our prompts, the names of the objects (e.g. blocks, trucks), predicates (e.g. on-table, in-city) and actions (e.g. pickup, drive) still do provide connections to the commonsense knowledge that the pretrained LLMs possess. One intriguing question is whether the planning performance is based really only on the domain model or patterns discovered from these other background connections. In order to test this, as part of our benchmark, we also provide obfuscated versions of the above domains, where the action names, predicate names and object names are obfuscated either with misleading words or random alphanumeric strings. Note that, for a standard planner, the original domain and the obfuscated version are identical. Further, we also provide the code to perform arbitrary obfuscations for the above domains or any additional domains that might be added in the future.

On the whole, our dataset consists of ∼26250 prompts across the various test cases and domains (including the obfuscated versions). We will now look at the results of GPT-4 and InstructGPT-3 on the Blocksworld domain in PlanBench.

# 7    Specimen Evaluation of PlanBench

While the objective of this paper is to make PlanBench available to other researchers, we also did some initial experiments to give useful baselines. Our evaluation here primarily focuses on two Large Language Models, GPT-4 and InstructGPT3. We used the OpenAI API to access these models. In particular, we evaluated the test framework on the Blocksworld domain. In Table 1, we have presented the results of GPT-4 and Instruct-GPT3 (text-davinci-002) on PlanBench. The best results within each model were observed for the auxiliary goal reformulation test cases. However, even the most effective model (GPT-4) falls short on most of the test cases in the Blocksworld domain of PlanBench. Overall, the performance of these LLMs on our benchmark shows that, as of right now, LLMs are pretty ineffective in reasoning about actions and change. PlanBench can thus serve as a useful marker of progress of LLMs in planning and reasoning. In a companion study [33], we conducted a deeper investigation into the planning abilities of current state-of-the-art LLMs, critically examining their plan generation abilities under various assumed roles. We performed experiments with different prompt configurations, delved into the failure modes of LLMs for autonomous plan generation and analyzed performance shifts when LLMs assume heuristic roles in planning systems.

# 8    Conclusion and Future Work

In this paper, we presented PlanBench, a reasoning assessment suite for large language models (LLMs) that consists of various test cases each evaluating a central aspect of planning and reasoning about actions and change. Our results show that even in simple common-sense planning domains, LLMs seem to display subpar performance. Our goal is to establish an extensible benchmark where researchers can evaluate the current and future large language models. Our assessment suite can be improved in multiple ways in the future. For instance, evaluation metrics that consider partial

Table 1: PlanBench Results of GPT-4 and Instruct-GPT3 (text-davinci-002) on Blocksworld domain. The tasks in the highlighted rows correspond to actual planning problems while the others correspond to simpler auxiliary planning tasks.

| Task | Instances correct | |
| --- | --- | --- |
| | GPT-4 | I-GPT3 |
| **Plan Generation** 
 We showcase an instance and the respective plan as an example and prompt the machine with a new instance. | 206/600 (34.3%) | 41/600 (6.8%) |
| **Cost-Optimal Planning** 
 We showcase an instance, the respective optimal plan and the associated cost as an example and prompt the machine with a new instance. | 198/600 (33%) | 35/600 (5.8%) |
| **Plan Verification** 
 We showcase three instances and three distinct plans (goal reaching, non goal-reaching and inexecutable) and present the respective validation and explanations. We then present a new instance and a plan and ask the machine for to verify and provide an explanation, if needed. | 352/600 (58.6%) | 72/600 (12%) |
| **Reasoning About Plan Execution** 
 We showcase an instance, an action sequence and the corresponding resulting state after executing the action sequence as an example. We then provide an instance and an executable action sequence and ask the machine to provide the resulting state. | 191/600 (31.8%) | 4/600 (0.6%) |
| **Replanning** 
 We showcase an instance, the respective plan and present an unexpected change of the state. We then also present a new plan from the changed state. Finally, for a new instance we repeat the same except we ask the machine for the new plan. | 289/600 (48.1%) | 40/600 (6.6%) |
| **Plan Generalization** 
 We showcase an instance and the respective plan as an example and prompt the machine with a new instance. The plans for both the instances can be generated by a fixed program containing loops and conditionals. | 141/500 (28.2%) | 49/500 (9.8%) |
| **Plan Reuse** 
 We showcase an instance and the respective plan as an example and prompt the machine with an instance which requires only a certain prefix of the plan provided in the example. | 392/600 (65.3%) | 102/600 (17%) |
| **Robustness to Goal Reformulation (Shuffling goal predicates)** 
 We showcase an instance and the respective plan as an example and prompt the machine with the same instance but shuffle the ordering of the goals. | 461/600 (76.8%) | 467/600 (77.8%) |
| **Robustness to Goal Reformulation (Full → Partial)** 
 We showcase an instance with a fully specified goal state and the respective plan as an example and prompt the machine with the same instance but provide a partially specified goal state. | 522/600 (87%) | 467/600 (77.8%) |
| **Robustness to Goal Reformulation (Partial → Full)** 
 We showcase an instance with a partially specified goal state and the respective plan as an example and prompt the machine with the same instance but provide a fully specified goal state. | 348/600 (58%) | 363/600 (60.5%) |

correctness could be incorporated into the benchmark. Additionally, the benchmark could be extended to support more number of IPC domains. In conclusion, we hope that PlanBench encourages other researchers to test the capabilities of their systems across different LLM models [2, 4, 28, 36, 24, 32, 9] and even those that are finetuned for such tasks.

# 9   Acknowledgements

This research was supported by ONR grants N00014-18-1-2442, N00014-18-1-2840, N00014-19-1-2119 and N00014-23-1-2409, AFOSR grant FA9550-18-1-0067, DARPA SAIL-ON grant W911NF-19-2-0006, and a JP Morgan AI Faculty Research Grant to Kambhampati. Sreedharan was supported in part by NSF grant 2303019. We would also like to thank Kaya Stechly and Anil Murthy for their help in adding additional domains to the benchmark.

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
