# A  Appendix

# Contents

## A.1 Broader Impact

PlanBench seeks to cover a broad range of tasks related to planning and reasoning. Our benchmark serves as a useful marker of progress of LLMs in planning and reasoning and enables efficient comparisons of performance between LLMs in such tasks. As for ethical concerns, PlanBench does not contain any personally-identifiable or privacy-related information. While PlanBench is intended to measure planning capabilities, there is no guarantee that deploying models for external planning, based solely on their performance on this benchmark will result in correct or safe plans.

## A.2 Additional details on experiments

All the experiments were run using the OpenAI API with all default parameters except the temperature. The temperature was made 0, making the LLMs deterministic. For GPT-4, the version we used had an 8k context window and was used between the months of March and June. We used the Fast-Downward system [8] as the planner and VAL [10] as the plan validator in our framework.

## A.3 Obfuscation experiments

As shown in Table 2, the performance of both GPT-4 and Instruct-GPT3 decreases significantly when the domain is obfuscated either deceptively or randomly. This shows that whatever planning

Table 2: Results of GPT-4 and Instruct-GPT3 for the Plan Generation test case.

| Domain | Instances correct | |
|---|---|---|
| | **GPT-4** | **InstructGPT-3** |
| **Mystery Blocksworld (Deceptive)** | 26/600 (4.3%) | 14/600 (0.23%) |
| **Mystery Blocksworld (Randomized)** | 12/600 (2%) | 6/600 (1%) |

performance they showed in the Blocksworld domain was more likely due to pattern matching rather than reasoning (which should have been robust to such kinds of obfuscation).

## A.4 Analysis of GPT-4 plans

### A.4.1 Comparison to the dataset

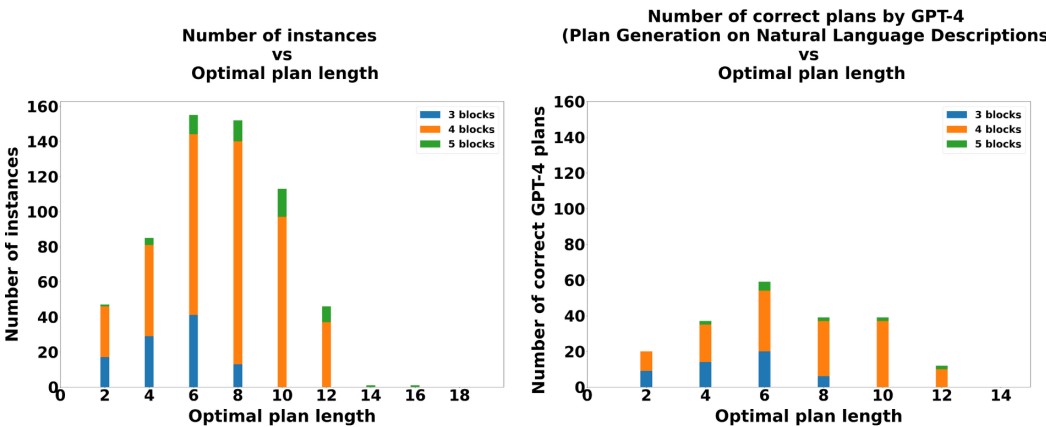

Figure 4: The graph on the left shows the distribution of the number of instances in the Blocksworld domain over their optimal plan lengths. The graph on the right shows the distribution of the number of correct plans by GPT-4 over optimal plan lengths.

Figure 4 provides insight into how the optimal plan length and block count distributions of Blocksworld instances where GPT-4 provides a correct plan stacks up against the optimal plan length and block count distributions of all instances. By comparing the two figures, we notice that more "shallow" instances - instances with a smaller optimal plan length - do not necessarily see an increase in the performance of LLMs. While perhaps deviating from intuition for classical planners, this is not unexpected given the knowledge of how LLMs work: regardless of the specifics of an instance, LLMs will attempt to solve it by predicting the next token based on the weights and the context.

### A.4.2 Failure modes analysis

To delve deeper into the failure cases of GPT-4, we performed a more forgiving evaluation of the validity of the generated plans in the plan generation test case. We used the concept of domain model relaxations from the automated planning community (which are used to derive heuristics [6]) and considered two types of relaxations: (i) *delete relaxation* which involves ignoring the delete conditions of the domain actions and (ii) *precondition relaxation* which ignores all the preconditions of the actions–thus assuming that any action is executable at any state giving their effects. We evaluated GPT-4's plans with respect to domain models that are delete-relaxed, precondition-relaxed or both. Note that a plan that is correct in the unrelaxed model will also be correct in all the relaxed versions. We observe (as shown in Figure 5) that even in the most relaxed version of the evaluation, GPT-4 still fails in a lot of instances.

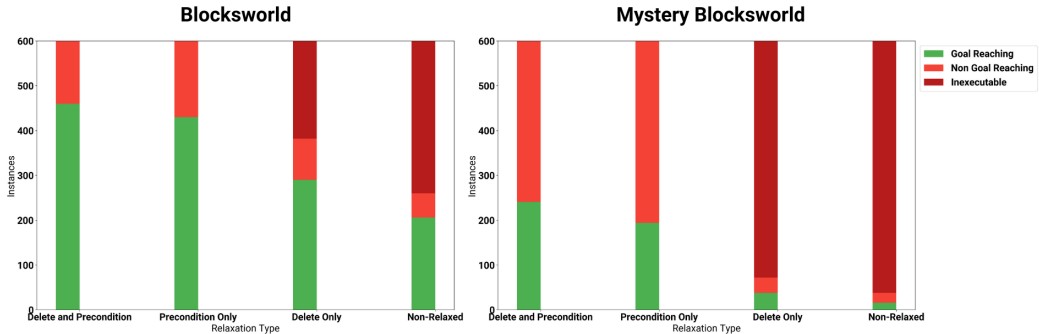

Figure 5: GPT-4 failure cases in the plan generation test case

## A.5  Plan Generation Prompts

### A.5.1  Blocksworld

---

**Blocksworld Plan Generation Prompt with GPT-4 Completion**

```
I am playing with a set of blocks where I need to arrange the blocks into stacks. Here are the
↪  actions I can do

Pick up a block
Unstack a block from on top of another block
Put down a block
Stack a block on top of another block

I have the following restrictions on my actions:
I can only pick up or unstack one block at a time.
I can only pick up or unstack a block if my hand is empty.
I can only pick up a block if the block is on the table and the block is clear. A block is clear
↪  if the block has no other blocks on top of it and if the block is not picked up.
I can only unstack a block from on top of another block if the block I am unstacking was really on
↪  top of the other block.
I can only unstack a block from on top of another block if the block I am unstacking is clear.
Once I pick up or unstack a block, I am holding the block.
I can only put down a block that I am holding.
I can only stack a block on top of another block if I am holding the block being stacked.
I can only stack a block on top of another block if the block onto which I am stacking the block
↪  is clear.
Once I put down or stack a block, my hand becomes empty.

[STATEMENT]
As initial conditions I have that, the red block is clear, the blue block is clear, the yellow
↪  block is clear, the hand is empty, the blue block is on top of the orange block, the red block
↪  is on the table, the orange block is on the table and the yellow block is on the table.
My goal is to have that the orange block is on top of the blue block.

My plan is as follows:

[PLAN]
unstack the blue block from on top of the orange block
put down the blue block
pick up the orange block
stack the orange block on top of the blue block
[PLAN END]

[STATEMENT]
As initial conditions I have that, the red block is clear, the yellow block is clear, the hand is
↪  empty, the red block is on top of the blue block, the yellow block is on top of the orange
↪  block, the blue block is on the table and the orange block is on the table.
My goal is to have that the orange block is on top of the red block.

My plan is as follows:

[PLAN]

------------GPT-4 response----------------
```

---

```
unstack the yellow block from on top of the orange block
put down the yellow block
unstack the red block from on top of the blue block
put down the red block
pick up the orange block
stack the orange block on top of the red block
[PLAN END]

------------Extracted LLM plan----------------
(unstack d c)
(put-down d)
(unstack a b)
(put-down a)
(pick-up c)
(stack c a)
------------Ground truth plan----------------
(unstack d c)
(put-down d)
(pick-up c)
(stack c a)
===================SUCCESS===================
```

## A.5.2 Logistics

### Logistics Plan Generation Prompt with GPT-4 Completion

```
I have to plan logistics to transport packages within cities via trucks and between cities via
↪  airplanes. Locations within a city are directly connected (trucks can move between any two
↪  such locations), and so are the cities. In each city there is exactly one truck and each city
↪  has one location that serves as an airport.
Here are the actions that can be performed:

Load a package into a truck. For example, load package_1 into truck_1 at location_1_1.
Load a package into an airplane. For example, load package_1 into airplane_1 at location_1_1.
Unload a package from a truck. For example, unload package_1 from truck_1 at location_1_1.
Unload a package from an airplane. For example, unload package_1 from airplane_1 at location_1_1.
Drive a truck from one location to another location. For example, drive truck_1 from location_1_1
↪  to location_1_2 in city_1.
Fly an airplane from one city to another city. For example, fly airplane_1 from location_1_1 to
↪  location_2_1. Here location_1_1 is the airport in city_1 and location_2_1 is the airport in
↪  city_2.

The following are the restrictions on the actions:
A package can be loaded into a truck only if the package and the truck are in the same location.
Once a package is loaded into a truck, the package is not at the location and is in the truck.
A package can be loaded into an airplane only if the package and the airplane are in the same
↪  location.
Once a package is loaded into an airplane, the package is not at the location and is in the
↪  airplane.
A package can be unloaded from a truck only if the package is in the truck.
Once a package is unloaded from a truck, the package is not in the truck and is at the location of
↪  the truck.
A package can be unloaded from an airplane only if the package in the airplane.
Once a package is unloaded from an airplane, the package is not in the airplane and is at the
↪  location of the airplane.
A truck can be driven from one location to another if the truck is at the from-location and both
↪  from-location and to-location are locations in the same city.
Once a truck is driven from one location to another, it is not at the from-location and is at the
↪  to-location.
An airplane can be flown from one city to another if the from-location and the to-location are
↪  airports and the airplane is at the from-location.
Once an airplane is flown from one city to another the airplane is not at the from-location and is
↪  at the to-location.

[STATEMENT]
As initial conditions I have that, location_0_0 is an airport, location_1_0 is an airport,
↪  airplane_0 is at location_0_0, airplane_1 is at location_0_0, package_0 is at location_1_0,
↪  truck_0 is at location_0_0, truck_1 is at location_1_0, location_0_0 is in the city city_0 and
↪  location_1_0 is in the city city_1.
My goal is to have that package_0 is at location_0_0.

My plan is as follows:

[PLAN]
fly airplane_1 from location_0_0 to location_1_0
```

```
load package_0 into airplane_1 at location_1_0
fly airplane_1 from location_1_0 to location_0_0
unload package_0 from airplane_1 at location_0_0
[PLAN END]

[STATEMENT]
As initial conditions I have that, location_0_0 is an airport, location_1_0 is an airport,
↪  airplane_0 is at location_1_0, airplane_1 is at location_1_0, package_0 is at location_1_0,
↪  truck_0 is at location_0_0, truck_1 is at location_1_0, location_0_0 is in the city city_0 and
↪  location_1_0 is in the city city_1.
My goal is to have that package_0 is at location_0_0.

My plan is as follows:

[PLAN]
------------Ground truth plan----------------
(load-airplane p0 a1 l1-0)
(fly-airplane a1 l1-0 l0-0)
(unload-airplane p0 a1 l0-0)
```

### A.5.3 Mystery Blocksworld

Mystery Blocksworld Plan Generation Prompt with GPT-4 Completion

```
I am playing with a set of objects. Here are the actions I can do

    Attack object
    Feast object from another object
    Succumb object
    Overcome object from another object

I have the following restrictions on my actions:
    To perform Attack action, the following facts need to be true: Province object, Planet object,
    ↪  Harmony.
    Once Attack action is performed the following facts will be true: Pain object.
    Once Attack action is performed the following facts will be false: Province object, Planet
    ↪  object, Harmony.
    To perform Succumb action, the following facts need to be true: Pain object.
    Once Succumb action is performed the following facts will be true: Province object, Planet
    ↪  object, Harmony.
    Once Succumb action is performed the following facts will be false: Pain object.
    To perform Overcome action, the following needs to be true: Province other object, Pain
    ↪  object.
    Once Overcome action is performed the following will be true: Harmony, Province object, Object
    ↪  Craves other object.
    Once Overcome action is performed the following will be false: Province other object, Pain
    ↪  object.
    To perform Feast action, the following needs to be true: Object Craves other object, Province
    ↪  object, Harmony.
    Once Feast action is performed the following will be true: Pain object, Province other object.
    Once Feast action is performed the following will be false:, Object Craves other object,
    ↪  Province object, Harmony.

[STATEMENT]
As initial conditions I have that, object b craves object c, harmony, planet object a, planet
↪  object c, planet object d, province object a, province object b and province object d.
My goal is to have that object c craves object b.

My plan is as follows:

[PLAN]
feast object b from object c
succumb object b
attack object c
overcome object c from object b
[PLAN END]

[STATEMENT]
As initial conditions I have that, object a craves object b, object d craves object c, harmony,
↪  planet object b, planet object c, province object a and province object d.
My goal is to have that object c craves object a.

My plan is as follows:

[PLAN]
```

```
------------Ground truth plan----------------
(feast object d object c)
(succumb object d)
(attack object c)
(overcome object c object a)
```

## A.6 Cost Optimal Planning Prompts

### A.6.1 Blocksworld

---

**Blocksworld Cost Optimal Planning Prompt with GPT-4 Completion**

```
I am playing with a set of blocks where I need to arrange the blocks into stacks. Here are the
↪   actions I can do:

Pick up a block. It takes 1 minute to pick up a block.
Unstack a block from on top of another block. It takes 1 minute to unstack a block from on top of
↪   another block.
Put down a block. It takes 1 minute to put down a block.
Stack a block on top of another block. It takes 1 minute to stack a block on top of another block.

I have the following restrictions on my actions:
I can only pick up or unstack one block at a time.
I can only pick up or unstack a block if my hand is empty.
I can only pick up a block if the block is on the table and the block is clear. A block is clear
↪   if the block has no other blocks on top of it and if the block is not picked up.
I can only unstack a block from on top of another block if the block I am unstacking was really on
↪   top of the other block.
I can only unstack a block from on top of another block if the block I am unstacking is clear.
Once I pick up or unstack a block, I am holding the block.
I can only put down a block that I am holding.
I can only stack a block on top of another block if I am holding the block being stacked.
I can only stack a block on top of another block if the block onto which I am stacking the block
↪   is clear.
Once I put down or stack a block, my hand becomes empty.
Once you stack a block on top of a second block, the second block is no longer clear.

[STATEMENT]
As initial conditions I have that, the red block is clear, the blue block is clear, the yellow
↪   block is clear, the hand is empty, the blue block is on top of the orange block, the red block
↪   is on the table, the orange block is on the table and the yellow block is on the table.
My goal is to have that the orange block is on top of the blue block. I want to minimize the time
↪   taken to achieve my goal.
My plan is as follows:

[PLAN]
unstack the blue block from on top of the orange block
put down the blue block
pick up the orange block
stack the orange block on top of the blue block
[PLAN END]
The total time to execute the plan is 4 minutes.

[STATEMENT]
As initial conditions I have that, the red block is clear, the yellow block is clear, the hand is
↪   empty, the red block is on top of the blue block, the yellow block is on top of the orange
↪   block, the blue block is on the table and the orange block is on the table.
My goal is to have that the orange block is on top of the red block. I want to minimize the time
↪   taken to achieve my goal.
My plan is as follows:

[PLAN]
------------GPT-4 response----------------
unstack the yellow block from on top of the orange block
put down the yellow block
unstack the red block from on top of the blue block
put down the red block
pick up the orange block
stack the orange block on top of the red block
[PLAN END]
The total time to execute the plan is 6 minutes.
------------Extracted LLM plan----------------
(unstack yellow orange)
(put-down yellow)
(unstack red blue)
```

---

```
(put-down red)
(pick-up orange)
(stack orange red)

-----------Ground truth plan----------------
(unstack yellow orange)
(put-down yellow)
(pick-up orange)
(stack orange red)

====================FAILURE====================
```

## A.6.2   Logistics

---

### Logistics Cost Optimal Planning Prompt with GPT-4 Completion

```
I have to plan logistics to transport packages within cities via trucks and between cities via
↪   airplanes. Locations within a city are directly connected (trucks can move between any two
↪   such locations), and so are the cities. In each city there is exactly one truck and each city
↪   has one location that serves as an airport.
Here are the actions that can be performed:

Load a package into a truck. It takes one minute to do this action.
Load a package into an airplane. It takes one minute to do this action.
Unload a package from a truck. It takes one minute to do this action.
Unload a package from an airplane. It takes one minute to do this action.
Drive a truck from one location to another location. It takes two minutes to do this action.
Fly an airplane from one city to another city. It takes five minutes to do this action.

The following are the restrictions on the actions:
A package can be loaded into a truck only if the package and the truck are in the same location.
Once a package is loaded into a truck, the package is not at the location and is in the truck.
A package can be loaded into an airplane only if the package and the airplane are in the same
↪   location.
Once a package is loaded into an airplane, the package is not at the location and is in the
↪   airplane.
A package can be unloaded from a truck only if the package is in the truck.
Once a package is unloaded from a truck, the package is not in the truck and is at the location of
↪   the truck.
A package can be unloaded from an airplane only if the package in the airplane.
Once a package is unloaded from an airplane, the package is not in the airplane and is at the
↪   location of the airplane.
A truck can be driven from one location to another if the truck is at the from-location and both
↪   from-location and to-location are locations in the same city.
Once a truck is driven from one location to another, it is not at the from-location and is at the
↪   to-location.
An airplane can be flown from one city to another if the from-location and the to-location are
↪   airports and the airplane is at the from-location.
Once an airplane is flown from one city to another the airplane is not at the from-location and is
↪   at the to-location.

[STATEMENT]
As initial conditions I have that, location_0_0 is an airport, location_1_0 is an airport,
↪   airplane_0 is at location_0_0, airplane_1 is at location_0_0, package_0 is at location_1_0,
↪   truck_0 is at location_0_0, truck_1 is at location_1_0, location_0_0 is in the city city_0 and
↪   location_1_0 is in the city city_1.
My goal is to have that package_0 is at location_0_0. I want to minimize the time taken to achieve
↪   my goal.
My plan is as follows:

[PLAN]
fly airplane_1 from location_0_0 to location_1_0
load package_0 into airplane_1 at location_1_0
fly airplane_1 from location_1_0 to location_0_0
unload package_0 from airplane_1 at location_0_0
[PLAN END]
The total time to execute the plan is 4 minutes.

[STATEMENT]
As initial conditions I have that, location_0_0 is an airport, location_1_0 is an airport,
↪   airplane_0 is at location_1_0, airplane_1 is at location_1_0, package_0 is at location_1_0,
↪   truck_0 is at location_0_0, truck_1 is at location_1_0, location_0_0 is in the city city_0 and
↪   location_1_0 is in the city city_1.
My goal is to have that package_0 is at location_0_0. I want to minimize the time taken to achieve
↪   my goal.
```

---

```
My plan is as follows:

[PLAN]
------------Ground truth plan----------------
(load-airplane p0 a1 l1-0)
(fly-airplane a1 l1-0 l0-0)
(unload-airplane p0 a1 l0-0)
```

### A.6.3 Mystery Blocksworld

---

**Mystery Blocksworld Cost Optimal Planning Prompt with GPT-4 Completion**

```
I am playing with a set of objects. Here are the actions I can do

Attack object. It takes 1 minute to do the Attack action.
Feast object from another object. It takes 1 minute to do the Feast action.
Succumb object. It takes 1 minute to do the Succumb action.
Overcome object from another object. It takes 1 minute to do the Overcome action.

I have the following restrictions on my actions:
To perform Attack action, the following facts need to be true: Province object, Planet object,
↪   Harmony.
Once Attack action is performed the following facts will be true: Pain object.
Once Attack action is performed the following facts will be false: Province object, Planet object,
↪   Harmony.
To perform Succumb action, the following facts need to be true: Pain object.
Once Succumb action is performed the following facts will be true: Province object, Planet object,
↪   Harmony.
Once Succumb action is performed the following facts will be false: Pain object.
To perform Overcome action, the following needs to be true: Province other object, Pain object.
Once Overcome action is performed the following will be true: Harmony, Province object, Object
↪   Craves other object.
Once Overcome action is performed the following will be false: Province other object, Pain object.
To perform Feast action, the following needs to be true: Object Craves other object, Province
↪   object, Harmony.
Once Feast action is performed the following will be true: Pain object, Province other object.
Once Feast action is performed the following will be false:, Object Craves other object, Province
↪   object, Harmony.

[STATEMENT]
As initial conditions I have that, object b craves object c, harmony, planet object a, planet
↪   object c, planet object d, province object a, province object b and province object d.
My goal is to have that object c craves object b. I want to minimize the time taken to achieve my
↪   goal.
My plan is as follows:

[PLAN]
feast object b from object c
succumb object b
attack object c
overcome object c from object b
[PLAN END]
The total time to execute the plan is 4 minutes.

[STATEMENT]
As initial conditions I have that, object a craves object b, object d craves object c, harmony,
↪   planet object b, planet object c, province object a and province object d.
My goal is to have that object c craves object a. I want to minimize the time taken to achieve my
↪   goal.
My plan is as follows:

[PLAN]
------------Ground truth plan----------------
(feast object d object c)
(succumb object d)
(attack object c)
(overcome object c object a)
```

---

## A.7 Plan Verification Promptss

### A.7.1 Blocksworld

---

**Blocksworld Plan Verification Prompt with GPT-4 Completion**

```
I am playing with a set of blocks where I need to arrange the blocks into stacks. Here are the
↪  actions I can do

Pick up a block
Unstack a block from on top of another block
Put down a block
Stack a block on top of another block

I have the following restrictions on my actions:
I can only pick up or unstack one block at a time.
I can only pick up or unstack a block if my hand is empty.
I can only pick up a block if the block is on the table and the block is clear. A block is clear
↪  if the block has no other blocks on top of it and if the block is not picked up.
I can only unstack a block from on top of another block if the block I am unstacking was really on
↪  top of the other block.
I can only unstack a block from on top of another block if the block I am unstacking is clear.
Once I pick up or unstack a block, I am holding the block.
I can only put down a block that I am holding.
I can only stack a block on top of another block if I am holding the block being stacked.
I can only stack a block on top of another block if the block onto which I am stacking the block
↪  is clear.
Once I put down or stack a block, my hand becomes empty.
Once you stack a block on top of a second block, the second block is no longer clear.

[STATEMENT]
As initial conditions I have that, the blue block is clear, the orange block is clear, the yellow
↪  block is clear, the hand is empty, the blue block is on top of the red block, the red block is
↪  on the table, the orange block is on the table and the yellow block is on the table.
My goal is to have that the orange block is on top of the red block.
My plan is as follows:

[PLAN]
unstack the blue block from on top of the red block
put down the blue block
pick up the orange block
stack the orange block on top of the red block
[PLAN END]

[VERIFICATION]
The above plan is valid.

[STATEMENT]
As initial conditions I have that, the red block is clear, the orange block is clear, the white
↪  block is clear, the hand is empty, the blue block is on top of the yellow block, the white
↪  block is on top of the blue block, the red block is on the table, the orange block is on the
↪  table and the yellow block is on the table.
My goal is to have that the blue block is on top of the yellow block, the orange block is on top
↪  of the white block and the white block is on top of the red block.
My plan is as follows:

[PLAN]
unstack the white block from on top of the blue block
stack the white block on top of the red block
pick up the orange block
[PLAN END]

[VERIFICATION]
The above plan is invalid. This is the unmet goal condition:
the orange block is on top of the white block
[STATEMENT]
As initial conditions I have that, the yellow block is clear, the hand is empty, the red block is
↪  on top of the white block, the blue block is on top of the orange block, the yellow block is
↪  on top of the red block, the white block is on top of the blue block and the orange block is
↪  on the table.
My goal is to have that the yellow block is on top of the white block.
My plan is as follows:

[PLAN]
unstack the yellow block from on top of the red block
pick up the yellow block
put down the yellow block
put down the red block
```

---

```
stack the yellow block on top of the white block
[PLAN END]

[VERIFICATION]
The above plan is invalid. The following action at step 2 has unmet preconditions:
pick up the yellow block
The unmet preconditions are:
the yellow block is clear, the hand is empty and the yellow block is on the table
[STATEMENT]
As initial conditions I have that, the red block is clear, the yellow block is clear, the hand is
↪  empty, the red block is on top of the blue block, the yellow block is on top of the orange
↪  block, the blue block is on the table and the orange block is on the table.
My goal is to have that the orange block is on top of the red block.
My plan is as follows:

[PLAN]
unstack the yellow block from on top of the orange block
put down the yellow block
pick up the orange block
[PLAN END]

[VERIFICATION]
------------GPT-4 response----------------
The above plan is invalid. This is the unmet goal condition:
the orange block is on top of the red block
------------Ground truth plan----------------
The above plan is invalid. This is the unmet goal condition:
the orange block is on top of the red block
====================SUCCESS====================
```

## A.7.2  Logistics

Logistics Plan Verification Prompt with GPT-4 Completion

```
I have to plan logistics to transport packages within cities via trucks and between cities via
↪  airplanes. Locations within a city are directly connected (trucks can move between any two
↪  such locations), and so are the cities. In each city there is exactly one truck and each city
↪  has one location that serves as an airport.
Here are the actions that can be performed:

Load a package into a truck. For example, load package_1 into truck_1 at location_1_1.
Load a package into an airplane. For example, load package_1 into airplane_1 at location_1_1.
Unload a package from a truck. For example, unload package_1 from truck_1 at location_1_1.
Unload a package from an airplane. For example, unload package_1 from airplane_1 at location_1_1.
Drive a truck from one location to another location. For example, drive truck_1 from location_1_1
↪  to location_1_2 in city_1.
Fly an airplane from one city to another city. For example, fly airplane_1 from location_1_1 to
↪  location_2_1. Here location_1_1 is the airport in city_1 and location_2_1 is the airport in
↪  city_2.

The following are the restrictions on the actions:
A package can be loaded into a truck only if the package and the truck are in the same location.
Once a package is loaded into a truck, the package is not at the location and is in the truck.
A package can be loaded into an airplane only if the package and the airplane are in the same
↪  location.
Once a package is loaded into an airplane, the package is not at the location and is in the
↪  airplane.
A package can be unloaded from a truck only if the package is in the truck.
Once a package is unloaded from a truck, the package is not in the truck and is at the location of
↪  the truck.
A package can be unloaded from an airplane only if the package in the airplane.
Once a package is unloaded from an airplane, the package is not in the airplane and is at the
↪  location of the airplane.
A truck can be driven from one location to another if the truck is at the from-location and both
↪  from-location and to-location are locations in the same city.
Once a truck is driven from one location to another, it is not at the from-location and is at the
↪  to-location.
An airplane can be flown from one city to another if the from-location and the to-location are
↪  airports and the airplane is at the from-location.
Once an airplane is flown from one city to another the airplane is not at the from-location and is
↪  at the to-location.

[STATEMENT]
```

As initial conditions I have that, location_0_0 is an airport, location_1_0 is an airport,
↪  location_2_0 is an airport, airplane_0 is at location_1_0, airplane_1 is at location_2_0,
↪  package_0 is at location_2_0, package_1 is at location_1_0, truck_0 is at location_0_0,
↪  truck_1 is at location_1_0, truck_2 is at location_2_0, location_0_0 is in the city city_0,
↪  location_1_0 is in the city city_1 and location_2_0 is in the city city_2.
My goal is to have that package_0 is at location_2_0 and package_1 is at location_0_0.
My plan is as follows:

[PLAN]
load package_1 into airplane_0 at location_1_0
fly airplane_0 from location_1_0 to location_0_0
unload package_1 from airplane_0 at location_0_0
[PLAN END]

[VERIFICATION]
The above plan is valid.

[STATEMENT]
As initial conditions I have that, location_0_0 is an airport, location_1_0 is an airport,
↪  location_2_0 is an airport, airplane_0 is at location_2_0, airplane_1 is at location_1_0,
↪  airplane_2 is at location_2_0, package_0 is at location_2_0, package_1 is at location_0_1,
↪  package_2 is at location_1_0, package_3 is at location_2_1, package_4 is at location_1_1,
↪  truck_0 is at location_0_0, truck_1 is at location_1_1, truck_2 is at location_2_1,
↪  location_0_0 is in the city city_0, location_0_1 is in the city city_0, location_1_0 is in the
↪  city city_1, location_1_1 is in the city city_1, location_2_0 is in the city city_2 and
↪  location_2_1 is in the city city_2.
My goal is to have that package_0 is at location_2_1, package_1 is at location_0_0, package_2 is
↪  at location_0_1, package_3 is at location_1_1 and package_4 is at location_2_0.
My plan is as follows:

[PLAN]
load package_2 into airplane_1 at location_1_0
drive truck_0 from location_0_0 to location_0_1 in city_0
fly airplane_1 from location_1_0 to location_0_0
unload package_3 from truck_1 at location_1_1
drive truck_0 from location_0_1 to location_0_0 in city_0
unload package_1 from truck_0 at location_0_0
fly airplane_2 from location_2_0 to location_1_0
unload package_3 from airplane_2 at location_1_0
load package_1 into truck_0 at location_0_1
drive truck_1 from location_1_0 to location_1_1 in city_1
unload package_2 from truck_0 at location_0_1
unload package_4 from truck_1 at location_1_0
load package_0 into truck_2 at location_2_0
unload package_0 from truck_2 at location_2_1
load package_2 into truck_0 at location_0_0
unload package_4 from airplane_2 at location_2_0
load package_3 into airplane_2 at location_2_0
drive truck_2 from location_2_0 to location_2_1 in city_2
drive truck_2 from location_2_1 to location_2_0 in city_2
unload package_2 from airplane_1 at location_0_0
unload package_3 from truck_2 at location_2_0
load package_3 into truck_2 at location_2_1
load package_4 into airplane_2 at location_1_0
load package_4 into truck_1 at location_1_1
drive truck_1 from location_1_1 to location_1_0 in city_1
fly airplane_2 from location_1_0 to location_2_0
[PLAN END]

[VERIFICATION]
The above plan is invalid.The following action at step 4 has an unmet precondition:
unload package_3 from truck_1 at location_1_1
The unmet precondition is:
package_3 is in truck_1
[STATEMENT]
As initial conditions I have that, location_0_0 is an airport, location_1_0 is an airport,
↪  location_2_0 is an airport, airplane_0 is at location_2_0, airplane_1 is at location_2_0,
↪  package_0 is at location_0_1, package_1 is at location_1_1, package_2 is at location_1_1,
↪  package_3 is at location_2_1, package_4 is at location_1_0, package_5 is at location_2_1,
↪  truck_0 is at location_0_1, truck_1 is at location_1_1, truck_2 is at location_2_1,
↪  location_0_0 is in the city city_0, location_0_1 is in the city city_0, location_1_0 is in the
↪  city city_1, location_1_1 is in the city city_1, location_2_0 is in the city city_2 and
↪  location_2_1 is in the city city_2.
My goal is to have that package_0 is at location_1_1, package_1 is at location_2_0, package_2 is
↪  at location_1_0, package_3 is at location_1_0, package_4 is at location_1_1 and package_5 is
↪  at location_1_0.
My plan is as follows:

```
[PLAN]
load package_5 into truck_2 at location_2_1
load package_3 into truck_2 at location_2_1
drive truck_2 from location_2_1 to location_2_0 in city_2
unload package_5 from truck_2 at location_2_0
unload package_3 from truck_2 at location_2_0
load package_2 into truck_1 at location_1_1
load package_1 into truck_1 at location_1_1
load package_0 into truck_0 at location_0_1
drive truck_0 from location_0_1 to location_0_0 in city_0
unload package_0 from truck_0 at location_0_0
load package_5 into airplane_1 at location_2_0
load package_3 into airplane_1 at location_2_0
drive truck_1 from location_1_1 to location_1_0 in city_1
load package_4 into truck_1 at location_1_0
unload package_2 from truck_1 at location_1_0
fly airplane_1 from location_2_0 to location_0_0
load package_0 into airplane_1 at location_0_0
fly airplane_1 from location_0_0 to location_1_0
unload package_5 from airplane_1 at location_1_0
unload package_3 from airplane_1 at location_1_0
unload package_0 from airplane_1 at location_1_0
load package_0 into truck_1 at location_1_0
unload package_1 from truck_1 at location_1_0
drive truck_1 from location_1_0 to location_1_1 in city_1
unload package_4 from truck_1 at location_1_1
unload package_0 from truck_1 at location_1_1
load package_1 into airplane_1 at location_1_0
fly airplane_1 from location_1_0 to location_2_0
[PLAN END]

[VERIFICATION]
The above plan is invalid. This is the unmet goal condition:
package_1 is at location_2_0
[STATEMENT]
As initial conditions I have that, location_0_0 is an airport, location_1_0 is an airport,
↪  airplane_0 is at location_1_0, airplane_1 is at location_1_0, package_0 is at location_1_0,
↪  truck_0 is at location_0_0, truck_1 is at location_1_0, location_0_0 is in the city city_0 and
↪  location_1_0 is in the city city_1.
My goal is to have that package_0 is at location_0_0.
My plan is as follows:

[PLAN]
load package_0 into airplane_1 at location_1_0
unload package_0 from airplane_1 at location_0_0
[PLAN END]

[VERIFICATION]
------------Ground truth plan----------------
The above plan is invalid.The following action at step 2 has an unmet precondition:
unload package_0 from airplane_1 at location_0_0
The unmet precondition is:
airplane_1 is at location_0_0
```

### A.7.3  Mystery Blocksworld

```
I am playing with a set of objects. Here are the actions I can do

    Attack object
    Feast object from another object
    Succumb object
    Overcome object from another object

I have the following restrictions on my actions:
    To perform Attack action, the following facts need to be true: Province object, Planet object,
    ↪  Harmony.
    Once Attack action is performed the following facts will be true: Pain object.
    Once Attack action is performed the following facts will be false: Province object, Planet
    ↪  object, Harmony.
    To perform Succumb action, the following facts need to be true: Pain object.
    Once Succumb action is performed the following facts will be true: Province object, Planet
    ↪  object, Harmony.
    Once Succumb action is performed the following facts will be false: Pain object.
```

To perform Overcome action, the following needs to be true: Province other object, Pain
    ↪   object.
    Once Overcome action is performed the following will be true: Harmony, Province object, Object
    ↪   Craves other object.
    Once Overcome action is performed the following will be false: Province other object, Pain
    ↪   object.
    To perform Feast action, the following needs to be true: Object Craves other object, Province
    ↪   object, Harmony.
    Once Feast action is performed the following will be true: Pain object, Province other object.
    Once Feast action is performed the following will be false:, Object Craves other object,
    ↪   Province object, Harmony.

[STATEMENT]
As initial conditions I have that, object b craves object a, harmony, planet object a, planet
↪   object c, planet object d, province object b, province object c and province object d.
My goal is to have that object c craves object a.
My plan is as follows:

[PLAN]
feast object b from object a
succumb object b
attack object c
overcome object c from object a
[PLAN END]

[VERIFICATION]
The above plan is valid.

[STATEMENT]
As initial conditions I have that, object b craves object d, object e craves object b, harmony,
↪   planet object a, planet object c, planet object d, province object a, province object c and
↪   province object e.
My goal is to have that object b craves object d, object c craves object e and object e craves
↪   object a.
My plan is as follows:

[PLAN]
feast object e from object b
overcome object e from object a
attack object c
[PLAN END]

[VERIFICATION]
The above plan is invalid. This is the unmet goal condition:
object c craves object e
[STATEMENT]
As initial conditions I have that, object a craves object e, object b craves object c, object d
↪   craves object a, object e craves object b, harmony, planet object c and province object d.
My goal is to have that object d craves object e.
My plan is as follows:

[PLAN]
feast object d from object a
attack object d
succumb object d
succumb object a
overcome object d from object e
[PLAN END]

[VERIFICATION]
The above plan is invalid. The following action at step 2 has unmet preconditions:
attack object d
The unmet preconditions are:
harmony, planet object d and province object d
[STATEMENT]
As initial conditions I have that, object a craves object b, object d craves object c, harmony,
↪   planet object b, planet object c, province object a and province object d.
My goal is to have that object c craves object a.
My plan is as follows:

[PLAN]
feast object d from object c
succumb object d
attack object c
[PLAN END]

[VERIFICATION]
------------Ground truth plan----------------

## A.8 Reasoning About Plan Execution Prompts

### A.8.1 Blocksworld

---

**Blocksworld Reasoning About Plan Execution Prompt with GPT-4 Completion**

```
I am playing with a set of blocks where I need to arrange the blocks into stacks. Here are the
↪  actions I can do

Pick up a block
Unstack a block from on top of another block
Put down a block
Stack a block on top of another block

I have the following restrictions on my actions:
I can only pick up or unstack one block at a time.
I can only pick up or unstack a block if my hand is empty.
I can only pick up a block if the block is on the table and the block is clear. A block is clear
↪  if the block has no other blocks on top of it and if the block is not picked up.
I can only unstack a block from on top of another block if the block I am unstacking was really on
↪  top of the other block.
I can only unstack a block from on top of another block if the block I am unstacking is clear.
Once I pick up or unstack a block, I am holding the block.
I can only put down a block that I am holding.
I can only stack a block on top of another block if I am holding the block being stacked.
I can only stack a block on top of another block if the block onto which I am stacking the block
↪  is clear.
Once I put down or stack a block, my hand becomes empty.
Once you stack a block on top of a second block, the second block is no longer clear.

[STATEMENT]
As initial conditions I have that, the red block is clear, the blue block is clear, the yellow
↪  block is clear, the hand is empty, the blue block is on top of the orange block, the red block
↪  is on the table, the orange block is on the table and the yellow block is on the table.
 I have executed the following action sequence:

[ACTION SEQUENCE]
unstack the blue block from on top of the orange block
put down the blue block
pick up the orange block
[ACTION SEQUENCE END]
[RESULTING STATE]
the red block is clear, the blue block is clear, the yellow block is clear, the hand is currently
↪  holding orange block, the red block is on the table, the blue block is on the table and the
↪  yellow block is on the table

[STATEMENT]
As initial conditions I have that, the red block is clear, the yellow block is clear, the hand is
↪  empty, the red block is on top of the blue block, the yellow block is on top of the orange
↪  block, the blue block is on the table and the orange block is on the table.
 I have executed the following action sequence:

[ACTION SEQUENCE]
unstack the yellow block from on top of the orange block
[ACTION SEQUENCE END]
[RESULTING STATE]

------------GPT-4 response----------------
the red block is clear, the yellow block is being held, the blue block is clear, the hand is
↪  currently holding the yellow block, the red block is on top of the blue block, the orange
↪  block is clear, the blue block is on the table, and the orange block is on the table.
------------Extracted LLM plan----------------
clear_a
clear_b
holding_d
on_a_b
clear_c
ontable_b
ontable_c
------------Ground truth plan----------------
holding_d
on_a_b
```

---

```
ontable_c
clear_a
ontable_b
clear_c
===================FAILURE====================
```

## A.8.2 Logistics

### Logistics Reasoning About Plan Execution Prompt with GPT-4 Completion

```
I have to plan logistics to transport packages within cities via trucks and between cities via
↪  airplanes. Locations within a city are directly connected (trucks can move between any two
↪  such locations), and so are the cities. In each city there is exactly one truck and each city
↪  has one location that serves as an airport.
Here are the actions that can be performed:

Load a package into a truck. For example, load package_1 into truck_1 at location_1_1.
Load a package into an airplane. For example, load package_1 into airplane_1 at location_1_1.
Unload a package from a truck. For example, unload package_1 from truck_1 at location_1_1.
Unload a package from an airplane. For example, unload package_1 from airplane_1 at location_1_1.
Drive a truck from one location to another location. For example, drive truck_1 from location_1_1
↪  to location_1_2 in city_1.
Fly an airplane from one city to another city. For example, fly airplane_1 from location_1_1 to
↪  location_2_1. Here location_1_1 is the airport in city_1 and location_2_1 is the airport in
↪  city_2.

The following are the restrictions on the actions:
A package can be loaded into a truck only if the package and the truck are in the same location.
Once a package is loaded into a truck, the package is not at the location and is in the truck.
A package can be loaded into an airplane only if the package and the airplane are in the same
↪  location.
Once a package is loaded into an airplane, the package is not at the location and is in the
↪  airplane.
A package can be unloaded from a truck only if the package is in the truck.
Once a package is unloaded from a truck, the package is not in the truck and is at the location of
↪  the truck.
A package can be unloaded from an airplane only if the package in the airplane.
Once a package is unloaded from an airplane, the package is not in the airplane and is at the
↪  location of the airplane.
A truck can be driven from one location to another if the truck is at the from-location and both
↪  from-location and to-location are locations in the same city.
Once a truck is driven from one location to another, it is not at the from-location and is at the
↪  to-location.
An airplane can be flown from one city to another if the from-location and the to-location are
↪  airports and the airplane is at the from-location.
Once an airplane is flown from one city to another the airplane is not at the from-location and is
↪  at the to-location.

[STATEMENT]
As initial conditions I have that, location_0_0 is an airport, location_1_0 is an airport,
↪  airplane_0 is at location_0_0, airplane_1 is at location_0_0, package_0 is at location_1_0,
↪  truck_0 is at location_0_0, truck_1 is at location_1_0, location_0_0 is in the city city_0 and
↪  location_1_0 is in the city city_1.
 I have executed the following action sequence:

[ACTION SEQUENCE]
fly airplane_1 from location_0_0 to location_1_0
load package_0 into airplane_1 at location_1_0
fly airplane_1 from location_1_0 to location_0_0
[ACTION SEQUENCE END]
[RESULTING STATE]
location_0_0 is an airport, location_1_0 is an airport, airplane_0 is at location_0_0, airplane_1
↪  is at location_0_0, truck_0 is at location_0_0, truck_1 is at location_1_0, location_0_0 is in
↪  the city city_0, location_1_0 is in the city city_1 and package_0 is in airplane_1

[STATEMENT]
As initial conditions I have that, location_0_0 is an airport, location_1_0 is an airport,
↪  airplane_0 is at location_1_0, airplane_1 is at location_1_0, package_0 is at location_1_0,
↪  truck_0 is at location_0_0, truck_1 is at location_1_0, location_0_0 is in the city city_0 and
↪  location_1_0 is in the city city_1.
 I have executed the following action sequence:

[ACTION SEQUENCE]
load package_0 into airplane_1 at location_1_0
[ACTION SEQUENCE END]
```

```
[RESULTING STATE]

------------Ground truth plan----------------
airplane_a1
truck_t0
city_c1
in-city_l1-0_c1
at_a0_l1-0
location_l0-0
obj_p0
airport_l0-0
at_t0_l0-0
truck_t1
location_l1-0
in-city_l0-0_c0
city_c0
at_a1_l1-0
airplane_a0
airport_l1-0
at_t1_l1-0
in_p0_a1
```

### A.8.3  Mystery Blocksworld

```
I am playing with a set of objects. Here are the actions I can do

    Attack object
    Feast object from another object
    Succumb object
    Overcome object from another object

I have the following restrictions on my actions:
    To perform Attack action, the following facts need to be true: Province object, Planet object,
    ↪  Harmony.
    Once Attack action is performed the following facts will be true: Pain object.
    Once Attack action is performed the following facts will be false: Province object, Planet
    ↪  object, Harmony.
    To perform Succumb action, the following facts need to be true: Pain object.
    Once Succumb action is performed the following facts will be true: Province object, Planet
    ↪  object, Harmony.
    Once Succumb action is performed the following facts will be false: Pain object.
    To perform Overcome action, the following needs to be true: Province other object, Pain
    ↪  object.
    Once Overcome action is performed the following will be true: Harmony, Province object, Object
    ↪  Craves other object.
    Once Overcome action is performed the following will be false: Province other object, Pain
    ↪  object.
    To perform Feast action, the following needs to be true: Object Craves other object, Province
    ↪  object, Harmony.
    Once Feast action is performed the following will be true: Pain object, Province other object.
    Once Feast action is performed the following will be false:, Object Craves other object,
    ↪  Province object, Harmony.

[STATEMENT]
As initial conditions I have that, object b craves object c, harmony, planet object a, planet
↪  object c, planet object d, province object a, province object b and province object d.
 I have executed the following action sequence:

[ACTION SEQUENCE]
feast object b from object c
succumb object b
attack object c
[ACTION SEQUENCE END]
[RESULTING STATE]
pain object c, planet object a, planet object b, planet object d, province object a, province
↪  object b and province object d

[STATEMENT]
As initial conditions I have that, object a craves object b, object d craves object c, harmony,
↪  planet object b, planet object c, province object a and province object d.
 I have executed the following action sequence:

[ACTION SEQUENCE]
```

```
feast object d from object c
[ACTION SEQUENCE END]
[RESULTING STATE]

------------Ground truth plan----------------
province_c
pain_d
province_a
planet_c
planet_b
craves_a_b
```

## A.9  Replanning Prompts

### A.9.1  Blocksworld

```
I am playing with a set of blocks where I need to arrange the blocks into stacks. Here are the
↪  actions I can do

Pick up a block
Unstack a block from on top of another block
Put down a block
Stack a block on top of another block

I have the following restrictions on my actions:
I can only pick up or unstack one block at a time.
I can only pick up or unstack a block if my hand is empty.
I can only pick up a block if the block is on the table and the block is clear. A block is clear
↪  if the block has no other blocks on top of it and if the block is not picked up.
I can only unstack a block from on top of another block if the block I am unstacking was really on
↪  top of the other block.
I can only unstack a block from on top of another block if the block I am unstacking is clear.
Once I pick up or unstack a block, I am holding the block.
I can only put down a block that I am holding.
I can only stack a block on top of another block if I am holding the block being stacked.
I can only stack a block on top of another block if the block onto which I am stacking the block
↪  is clear.
Once I put down or stack a block, my hand becomes empty.
Once you stack a block on top of a second block, the second block is no longer clear.

[STATEMENT]
As initial conditions I have that, the red block is clear, the blue block is clear, the yellow
↪  block is clear, the hand is empty, the blue block is on top of the orange block, the red block
↪  is on the table, the orange block is on the table and the yellow block is on the table.
My goal is to have that the orange block is on top of the blue block.
My plan is as follows:

[PLAN]
unstack the blue block from on top of the orange block
put down the blue block
pick up the orange block
stack the orange block on top of the blue block
[PLAN END]

During execution, an unexpected event has occurred.
After executing the action "pick up the orange block" at step 3 in the plan, the following facts
↪  unexpectedly became true: the orange block is clear, the hand is empty and the orange block is
↪  on top of the red block
The following facts became unexpectedly false: the red block is clear and the hand is currently
↪  holding orange block
After re-planning from the new state, the plan is as follows:
[PLAN]
unstack the orange block from on top of the red block
stack the orange block on top of the blue block
[PLAN END]

[STATEMENT]
As initial conditions I have that, the red block is clear, the yellow block is clear, the hand is
↪  empty, the red block is on top of the blue block, the yellow block is on top of the orange
↪  block, the blue block is on the table and the orange block is on the table.
My goal is to have that the orange block is on top of the red block.
My plan is as follows:
```

```
[PLAN]
unstack the yellow block from on top of the orange block
put down the yellow block
pick up the orange block
stack the orange block on top of the red block
[PLAN END]

During execution, an unexpected event has occurred.
After executing the action "pick up the orange block" at step 3 in the plan, the following facts
↪   unexpectedly became true: the orange block is clear, the hand is empty and the orange block is
↪   on top of the red block
The following facts became unexpectedly false: the red block is clear and the hand is currently
↪   holding orange block
After re-planning from the new state, the plan is as follows:
[PLAN]
------------GPT-4 response----------------
No further actions are needed as the goal has been achieved: the orange block is on top of the red
↪   block.
------------Extracted LLM plan----------------

------------Ground truth plan----------------

====================SUCCESS====================
```

## A.9.2 Logistics

### Logistics Replanning Prompt with GPT-4 Completion

```
I have to plan logistics to transport packages within cities via trucks and between cities via
↪   airplanes. Locations within a city are directly connected (trucks can move between any two
↪   such locations), and so are the cities. In each city there is exactly one truck and each city
↪   has one location that serves as an airport.
Here are the actions that can be performed:

Load a package into a truck. For example, load package_1 into truck_1 at location_1_1.
Load a package into an airplane. For example, load package_1 into airplane_1 at location_1_1.
Unload a package from a truck. For example, unload package_1 from truck_1 at location_1_1.
Unload a package from an airplane. For example, unload package_1 from airplane_1 at location_1_1.
Drive a truck from one location to another location. For example, drive truck_1 from location_1_1
↪   to location_1_2 in city_1.
Fly an airplane from one city to another city. For example, fly airplane_1 from location_1_1 to
↪   location_2_1. Here location_1_1 is the airport in city_1 and location_2_1 is the airport in
↪   city_2.

The following are the restrictions on the actions:
A package can be loaded into a truck only if the package and the truck are in the same location.
Once a package is loaded into a truck, the package is not at the location and is in the truck.
A package can be loaded into an airplane only if the package and the airplane are in the same
↪   location.
Once a package is loaded into an airplane, the package is not at the location and is in the
↪   airplane.
A package can be unloaded from a truck only if the package is in the truck.
Once a package is unloaded from a truck, the package is not in the truck and is at the location of
↪   the truck.
A package can be unloaded from an airplane only if the package in the airplane.
Once a package is unloaded from an airplane, the package is not in the airplane and is at the
↪   location of the airplane.
A truck can be driven from one location to another if the truck is at the from-location and both
↪   from-location and to-location are locations in the same city.
Once a truck is driven from one location to another, it is not at the from-location and is at the
↪   to-location.
An airplane can be flown from one city to another if the from-location and the to-location are
↪   airports and the airplane is at the from-location.
Once an airplane is flown from one city to another the airplane is not at the from-location and is
↪   at the to-location.

[STATEMENT]
As initial conditions I have that, location_0_0 is an airport, location_1_0 is an airport,
↪   airplane_0 is at location_0_0, airplane_1 is at location_0_0, package_0 is at location_1_0,
↪   truck_0 is at location_0_0, truck_1 is at location_1_0, location_0_0 is in the city city_0 and
↪   location_1_0 is in the city city_1.
My goal is to have that package_0 is at location_0_0.
My plan is as follows:

[PLAN]
```

```
fly airplane_1 from location_0_0 to location_1_0
load package_0 into airplane_1 at location_1_0
fly airplane_1 from location_1_0 to location_0_0
unload package_0 from airplane_1 at location_0_0
[PLAN END]

During execution, an unexpected event has occurred.
After executing the action "fly airplane_1 from location_1_0 to location_0_0" at step 3 in the
↪  plan, the following facts unexpectedly became true: package_0 is at location_1_0
The following facts became unexpectedly false: package_0 is in airplane_1
After re-planning from the new state, the plan is as follows:
[PLAN]
fly airplane_1 from location_0_0 to location_1_0
load package_0 into airplane_1 at location_1_0
fly airplane_1 from location_1_0 to location_0_0
unload package_0 from airplane_1 at location_0_0
[PLAN END]

[STATEMENT]
As initial conditions I have that, location_0_0 is an airport, location_1_0 is an airport,
↪  airplane_0 is at location_1_0, airplane_1 is at location_1_0, package_0 is at location_1_0,
↪  truck_0 is at location_0_0, truck_1 is at location_1_0, location_0_0 is in the city city_0 and
↪  location_1_0 is in the city city_1.
My goal is to have that package_0 is at location_0_0.
My plan is as follows:

[PLAN]
load package_0 into airplane_1 at location_1_0
fly airplane_1 from location_1_0 to location_0_0
unload package_0 from airplane_1 at location_0_0
[PLAN END]

During execution, an unexpected event has occurred.
After executing the action "load package_0 into airplane_1 at location_1_0" at step 1 in the plan,
↪  the following facts unexpectedly became true: package_0 is at location_1_0
The following facts became unexpectedly false: package_0 is in airplane_1
After re-planning from the new state, the plan is as follows:
[PLAN]
------------Ground truth plan----------------
(load-airplane p0 a1 l1-0)
(fly-airplane a1 l1-0 l0-0)
(unload-airplane p0 a1 l0-0)
```

### A.9.3 Mystery Blocksworld

```
I am playing with a set of objects. Here are the actions I can do

    Attack object
    Feast object from another object
    Succumb object
    Overcome object from another object

I have the following restrictions on my actions:
    To perform Attack action, the following facts need to be true: Province object, Planet object,
    ↪  Harmony.
    Once Attack action is performed the following facts will be true: Pain object.
    Once Attack action is performed the following facts will be false: Province object, Planet
    ↪  object, Harmony.
    To perform Succumb action, the following facts need to be true: Pain object.
    Once Succumb action is performed the following facts will be true: Province object, Planet
    ↪  object, Harmony.
    Once Succumb action is performed the following facts will be false: Pain object.
    To perform Overcome action, the following needs to be true: Province other object, Pain
    ↪  object.
    Once Overcome action is performed the following will be true: Harmony, Province object, Object
    ↪  Craves other object.
    Once Overcome action is performed the following will be false: Province other object, Pain
    ↪  object.
    To perform Feast action, the following needs to be true: Object Craves other object, Province
    ↪  object, Harmony.
    Once Feast action is performed the following will be true: Pain object, Province other object.
    Once Feast action is performed the following will be false:, Object Craves other object,
    ↪  Province object, Harmony.
```

```
[STATEMENT]
As initial conditions I have that, object b craves object c, harmony, planet object a, planet
↪ object c, planet object d, province object a, province object b and province object d.
My goal is to have that object c craves object b.
My plan is as follows:

[PLAN]
feast object b from object c
succumb object b
attack object c
overcome object c from object b
[PLAN END]

During execution, an unexpected event has occurred.
After executing the action "attack object c" at step 3 in the plan, the following facts
↪ unexpectedly became true: object c craves object a, harmony and province object c
The following facts became unexpectedly false: pain object c and province object a
After re-planning from the new state, the plan is as follows:
[PLAN]
feast object c from object a
overcome object c from object b
[PLAN END]

[STATEMENT]
As initial conditions I have that, object a craves object b, object d craves object c, harmony,
↪ planet object b, planet object c, province object a and province object d.
My goal is to have that object c craves object a.
My plan is as follows:

[PLAN]
feast object d from object c
succumb object d
attack object c
overcome object c from object a
[PLAN END]

During execution, an unexpected event has occurred.
After executing the action "attack object c" at step 3 in the plan, the following facts
↪ unexpectedly became true: object c craves object a, harmony and province object c
The following facts became unexpectedly false: pain object c and province object a
After re-planning from the new state, the plan is as follows:
[PLAN]
------------Ground truth plan----------------
```

## A.10   Plan Generalization Prompts

### A.10.1   Blocksworld

```
I am playing with a set of blocks where I need to arrange the blocks into stacks. Here are the
↪ actions I can do

Pick up a block
Unstack a block from on top of another block
Put down a block
Stack a block on top of another block

I have the following restrictions on my actions:
I can only pick up or unstack one block at a time.
I can only pick up or unstack a block if my hand is empty.
I can only pick up a block if the block is on the table and the block is clear. A block is clear
↪ if the block has no other blocks on top of it and if the block is not picked up.
I can only unstack a block from on top of another block if the block I am unstacking was really on
↪ top of the other block.
I can only unstack a block from on top of another block if the block I am unstacking is clear.
Once I pick up or unstack a block, I am holding the block.
I can only put down a block that I am holding.
I can only stack a block on top of another block if I am holding the block being stacked.
I can only stack a block on top of another block if the block onto which I am stacking the block
↪ is clear.
Once I put down or stack a block, my hand becomes empty.
Once you stack a block on top of a second block, the second block is no longer clear.
```

[STATEMENT]
As initial conditions I have that, the blue block is clear, the yellow block is clear, the magenta
↪  block is clear, the black block is clear, the cyan block is clear, the green block is clear,
↪  the violet block is clear, the gold block is clear, the hand is empty, the blue block is on
↪  the table, the yellow block is on the table, the magenta block is on the table, the black
↪  block is on the table, the cyan block is on the table, the green block is on the table, the
↪  violet block is on the table and the gold block is on the table.
My goal is to have that the blue block is on top of the cyan block, the yellow block is on top of
↪  the blue block, the magenta block is on top of the green block, the black block is on top of
↪  the yellow block, the cyan block is on top of the gold block, the green block is on top of the
↪  black block and the violet block is on top of the magenta block.

My plan is as follows:

[PLAN]
pick up the cyan block
stack the cyan block on top of the gold block
pick up the blue block
stack the blue block on top of the cyan block
pick up the yellow block
stack the yellow block on top of the blue block
pick up the black block
stack the black block on top of the yellow block
pick up the green block
stack the green block on top of the black block
pick up the magenta block
stack the magenta block on top of the green block
pick up the violet block
stack the violet block on top of the magenta block
[PLAN END]

[STATEMENT]
As initial conditions I have that, the red block is clear, the blue block is clear, the black
↪  block is clear, the cyan block is clear, the green block is clear, the violet block is clear,
↪  the silver block is clear, the hand is empty, the red block is on the table, the blue block is
↪  on the table, the black block is on the table, the cyan block is on the table, the green block
↪  is on the table, the violet block is on the table and the silver block is on the table.
My goal is to have that the red block is on top of the blue block, the blue block is on top of the
↪  silver block, the black block is on top of the red block, the cyan block is on top of the
↪  green block, the violet block is on top of the black block and the silver block is on top of
↪  the cyan block.

My plan is as follows:

[PLAN]
------------GPT-4 response----------------
pick up the blue block
stack the blue block on top of the silver block
pick up the red block
stack the red block on top of the blue block
pick up the black block
stack the black block on top of the red block
pick up the cyan block
stack the cyan block on top of the green block
pick up the silver block
stack the silver block on top of the cyan block
pick up the violet block
stack the violet block on top of the black block
[PLAN END]
------------Extracted LLM plan----------------
(pick-up blue)
(stack blue silver)
(pick-up red)
(stack red blue)
(pick-up black)
(stack black red)
(pick-up cyan)
(stack cyan green)
(pick-up silver)
(stack silver cyan)
(pick-up violet)
(stack violet black)

------------Ground truth plan----------------
(pick-up cyan)
(stack cyan green)
(pick-up silver)
(stack silver cyan)

```
(pick-up blue)
(stack blue silver)
(pick-up red)
(stack red blue)
(pick-up black)
(stack black red)
(pick-up violet)
(stack violet black)

===================FAILURE===================
```

## A.10.2   Logistics

Logistics Plan Generalization Prompt with GPT-4 Completion

```
I have to plan logistics to transport packages within cities via trucks and between cities via
↪   airplanes. Locations within a city are directly connected (trucks can move between any two
↪   such locations), and so are the cities. In each city there is exactly one truck and each city
↪   has one location that serves as an airport.
Here are the actions that can be performed:

Load a package into a truck. For example, load package_1 into truck_1 at location_1_1.
Load a package into an airplane. For example, load package_1 into airplane_1 at location_1_1.
Unload a package from a truck. For example, unload package_1 from truck_1 at location_1_1.
Unload a package from an airplane. For example, unload package_1 from airplane_1 at location_1_1.
Drive a truck from one location to another location. For example, drive truck_1 from location_1_1
↪   to location_1_2 in city_1.
Fly an airplane from one city to another city. For example, fly airplane_1 from location_1_1 to
↪   location_2_1. Here location_1_1 is the airport in city_1 and location_2_1 is the airport in
↪   city_2.

The following are the restrictions on the actions:
A package can be loaded into a truck only if the package and the truck are in the same location.
Once a package is loaded into a truck, the package is not at the location and is in the truck.
A package can be loaded into an airplane only if the package and the airplane are in the same
↪   location.
Once a package is loaded into an airplane, the package is not at the location and is in the
↪   airplane.
A package can be unloaded from a truck only if the package is in the truck.
Once a package is unloaded from a truck, the package is not in the truck and is at the location of
↪   the truck.
A package can be unloaded from an airplane only if the package in the airplane.
Once a package is unloaded from an airplane, the package is not in the airplane and is at the
↪   location of the airplane.
A truck can be driven from one location to another if the truck is at the from-location and both
↪   from-location and to-location are locations in the same city.
Once a truck is driven from one location to another, it is not at the from-location and is at the
↪   to-location.
An airplane can be flown from one city to another if the from-location and the to-location are
↪   airports and the airplane is at the from-location.
Once an airplane is flown from one city to another the airplane is not at the from-location and is
↪   at the to-location.

[STATEMENT]
As initial conditions I have that, location_0_0 is an airport, location_1_0 is an airport,
↪   location_2_0 is an airport, airplane_0 is at location_0_0, airplane_1 is at location_1_0,
↪   airplane_2 is at location_2_0, package_0 is at location_0_2, package_1 is at location_0_0,
↪   package_2 is at location_0_1, package_3 is at location_1_2, package_4 is at location_1_0,
↪   package_5 is at location_1_1, package_6 is at location_2_2, package_7 is at location_2_0,
↪   package_8 is at location_2_1, truck_0 is at location_0_2, truck_1 is at location_1_2, truck_2
↪   is at location_2_2, location_0_0 is in the city city_0, location_0_1 is in the city city_0,
↪   location_0_2 is in the city city_0, location_1_0 is in the city city_1, location_1_1 is in the
↪   city city_1, location_1_2 is in the city city_1, location_2_0 is in the city city_2,
↪   location_2_1 is in the city city_2 and location_2_2 is in the city city_2.
My goal is to have that package_0 is at location_0_1, package_1 is at location_2_0, package_2 is
↪   at location_0_0, package_3 is at location_1_1, package_4 is at location_2_0, package_5 is at
↪   location_1_0, package_6 is at location_2_1, package_7 is at location_1_0 and package_8 is at
↪   location_2_0.

My plan is as follows:

[PLAN]
load package_6 into truck_2 at location_2_2
drive truck_2 from location_2_2 to location_2_1 in city_2
load package_8 into truck_2 at location_2_1
```

```
unload package_6 from truck_2 at location_2_1
drive truck_2 from location_2_1 to location_2_0 in city_2
unload package_8 from truck_2 at location_2_0
load package_3 into truck_1 at location_1_2
drive truck_1 from location_1_2 to location_1_1 in city_1
load package_5 into truck_1 at location_1_1
unload package_3 from truck_1 at location_1_1
drive truck_1 from location_1_1 to location_1_0 in city_1
unload package_5 from truck_1 at location_1_0
load package_0 into truck_0 at location_0_2
load package_4 into airplane_1 at location_1_0
load package_1 into airplane_0 at location_0_0
drive truck_0 from location_0_2 to location_0_1 in city_0
load package_2 into truck_0 at location_0_1
unload package_0 from truck_0 at location_0_1
drive truck_0 from location_0_1 to location_0_0 in city_0
unload package_2 from truck_0 at location_0_0
fly airplane_0 from location_0_0 to location_2_0
unload package_1 from airplane_0 at location_2_0
load package_7 into airplane_2 at location_2_0
fly airplane_2 from location_2_0 to location_1_0
unload package_7 from airplane_2 at location_1_0
fly airplane_1 from location_1_0 to location_2_0
unload package_4 from airplane_1 at location_2_0
[PLAN END]

[STATEMENT]
As initial conditions I have that, location_0_5 is an airport, location_1_5 is an airport,
↪  location_2_5 is an airport, airplane_0 is at location_0_5, airplane_1 is at location_1_5,
↪  airplane_2 is at location_2_5, package_0 is at location_0_9, package_1 is at location_0_3,
↪  package_1 is at location_1_2, package_1 is at location_1_8, package_1 is at location_1_1,
↪  package_1 is at location_1_4, package_1 is at location_2_9, package_1 is at location_2_3,
↪  package_1 is at location_2_7, package_1 is at location_2_2, package_1 is at location_2_8,
↪  package_1 is at location_2_1, package_2 is at location_0_7, package_2 is at location_2_4,
↪  package_3 is at location_0_2, package_4 is at location_0_8, package_5 is at location_0_1,
↪  package_6 is at location_0_4, package_7 is at location_1_9, package_8 is at location_1_3,
↪  package_9 is at location_1_7, truck_0 is at location_0_8, truck_1 is at location_1_8, truck_2
↪  is at location_2_8, location_0_0 is in the city city_0, location_0_1 is in the city city_0,
↪  location_0_2 is in the city city_0, location_0_3 is in the city city_0, location_0_4 is in the
↪  city city_0, location_0_5 is in the city city_0, location_0_6 is in the city city_0,
↪  location_0_7 is in the city city_0, location_0_8 is in the city city_0, location_0_9 is in the
↪  city city_0, location_1_0 is in the city city_1, location_1_1 is in the city city_1,
↪  location_1_2 is in the city city_1, location_1_3 is in the city city_1, location_1_4 is in the
↪  city city_1, location_1_5 is in the city city_1, location_1_6 is in the city city_1,
↪  location_1_7 is in the city city_1, location_1_8 is in the city city_1, location_1_9 is in the
↪  city city_1, location_2_0 is in the city city_2, location_2_1 is in the city city_2,
↪  location_2_2 is in the city city_2, location_2_3 is in the city city_2, location_2_4 is in the
↪  city city_2, location_2_5 is in the city city_2, location_2_6 is in the city city_2,
↪  location_2_7 is in the city city_2, location_2_8 is in the city city_2 and location_2_9 is in
↪  the city city_2.
My goal is to have that package_0 is at location_0_2, package_1 is at location_0_9, package_1 is
↪  at location_1_4, package_1 is at location_1_3, package_1 is at location_1_7, package_1 is at
↪  location_1_1, package_1 is at location_2_2, package_1 is at location_2_9, package_1 is at
↪  location_1_5, package_1 is at location_2_4, package_1 is at location_2_3, package_1 is at
↪  location_2_7, package_2 is at location_2_5, package_2 is at location_2_1, package_3 is at
↪  location_0_4, package_4 is at location_0_3, package_5 is at location_0_7, package_6 is at
↪  location_0_1, package_7 is at location_1_2, package_8 is at location_1_9 and package_9 is at
↪  location_0_5.

My plan is as follows:

[PLAN]
------------Ground truth plan----------------
(load-truck p18 t2 l2-8)
(drive-truck t2 l2-8 l2-3 c2)
(unload-truck p18 t2 l2-3)
(load-truck p15 t2 l2-3)
(drive-truck t2 l2-3 l2-9 c2)
(unload-truck p15 t2 l2-9)
(load-truck p14 t2 l2-9)
(drive-truck t2 l2-9 l2-2 c2)
(load-truck p17 t2 l2-2)
(unload-truck p14 t2 l2-2)
(drive-truck t2 l2-2 l2-4 c2)
(load-truck p20 t2 l2-4)
(unload-truck p17 t2 l2-4)
(drive-truck t2 l2-4 l2-1 c2)
(unload-truck p20 t2 l2-1)
```

```
(load-truck p19 t2 l2-1)
(drive-truck t2 l2-1 l2-7 c2)
(unload-truck p19 t2 l2-7)
(load-truck p16 t2 l2-7)
(drive-truck t2 l2-7 l2-5 c2)
(unload-truck p16 t2 l2-5)
(load-truck p11 t1 l1-8)
(drive-truck t1 l1-8 l1-3 c1)
(load-truck p8 t1 l1-3)
(unload-truck p11 t1 l1-3)
(drive-truck t1 l1-3 l1-9 c1)
(unload-truck p8 t1 l1-9)
(load-truck p7 t1 l1-9)
(drive-truck t1 l1-9 l1-2 c1)
(unload-truck p7 t1 l1-2)
(load-truck p10 t1 l1-2)
(drive-truck t1 l1-2 l1-4 c1)
(load-truck p13 t1 l1-4)
(unload-truck p10 t1 l1-4)
(drive-truck t1 l1-4 l1-1 c1)
(unload-truck p13 t1 l1-1)
(load-truck p12 t1 l1-1)
(drive-truck t1 l1-1 l1-7 c1)
(load-truck p9 t1 l1-7)
(unload-truck p12 t1 l1-7)
(drive-truck t1 l1-7 l1-5 c1)
(load-truck p4 t0 l0-8)
(drive-truck t0 l0-8 l0-3 c0)
(unload-truck p4 t0 l0-3)
(load-truck p1 t0 l0-3)
(drive-truck t0 l0-3 l0-9 c0)
(unload-truck p1 t0 l0-9)
(load-truck p0 t0 l0-9)
(drive-truck t0 l0-9 l0-2 c0)
(load-truck p3 t0 l0-2)
(unload-truck p0 t0 l0-2)
(drive-truck t0 l0-2 l0-4 c0)
(load-truck p6 t0 l0-4)
(unload-truck p3 t0 l0-4)
(drive-truck t0 l0-4 l0-1 c0)
(unload-truck p6 t0 l0-1)
(load-truck p5 t0 l0-1)
(drive-truck t0 l0-1 l0-7 c0)
(unload-truck p5 t0 l0-7)
(load-truck p2 t0 l0-7)
(drive-truck t0 l0-7 l0-5 c0)
(unload-truck p2 t0 l0-5)
(load-airplane p2 a0 l0-5)
(unload-truck p9 t1 l1-5)
(load-airplane p16 a2 l2-5)
(fly-airplane a0 l0-5 l2-5)
(unload-airplane p2 a0 l2-5)
(fly-airplane a2 l2-5 l1-5)
(load-airplane p9 a2 l1-5)
(unload-airplane p16 a2 l1-5)
(fly-airplane a2 l1-5 l0-5)
(unload-airplane p9 a2 l0-5)
```

### A.10.3 Mystery Blocksworld

```
I am playing with a set of objects. Here are the actions I can do

    Attack object
    Feast object from another object
    Succumb object
    Overcome object from another object

I have the following restrictions on my actions:
    To perform Attack action, the following facts need to be true: Province object, Planet object,
    ↪  Harmony.
    Once Attack action is performed the following facts will be true: Pain object.
    Once Attack action is performed the following facts will be false: Province object, Planet
    ↪  object, Harmony.
```

To perform Succumb action, the following facts need to be true: Pain object.
        Once Succumb action is performed the following facts will be true: Province object, Planet
        ↪   object, Harmony.
        Once Succumb action is performed the following facts will be false: Pain object.
        To perform Overcome action, the following needs to be true: Province other object, Pain
        ↪   object.
        Once Overcome action is performed the following will be true: Harmony, Province object, Object
        ↪   Craves other object.
        Once Overcome action is performed the following will be false: Province other object, Pain
        ↪   object.
        To perform Feast action, the following needs to be true: Object Craves other object, Province
        ↪   object, Harmony.
        Once Feast action is performed the following will be true: Pain object, Province other object.
        Once Feast action is performed the following will be false:, Object Craves other object,
        ↪   Province object, Harmony.

[STATEMENT]
As initial conditions I have that, harmony, planet object b, planet object d, planet object f,
↪   planet object g, planet object h, planet object i, planet object j, planet object l, province
↪   object b, province object d, province object f, province object g, province object h, province
↪   object i, province object j and province object l.
My goal is to have that object b craves object h, object d craves object b, object f craves object
↪   i, object g craves object d, object h craves object l, object i craves object g and object j
↪   craves object f.

My plan is as follows:

[PLAN]
attack object h
overcome object h from object l
attack object b
overcome object b from object h
attack object d
overcome object d from object b
attack object g
overcome object g from object d
attack object i
overcome object i from object g
attack object f
overcome object f from object i
attack object j
overcome object j from object f
[PLAN END]

[STATEMENT]
As initial conditions I have that, harmony, planet object a, planet object b, planet object g,
↪   planet object h, planet object i, planet object j, planet object k, province object a,
↪   province object b, province object g, province object h, province object i, province object j
↪   and province object k.
My goal is to have that object a craves object b, object b craves object k, object g craves object
↪   a, object h craves object i, object j craves object g and object k craves object h.

My plan is as follows:

[PLAN]
------------Ground truth plan----------------
(attack h)
(overcome h i)
(attack k)
(overcome k h)
(attack b)
(overcome b k)
(attack a)
(overcome a b)
(attack g)
(overcome g a)
(attack j)
(overcome j g)

## A.11 Plan Reuse Prompts

### A.11.1 Blocksworld

---

**Blocksworld Plan Reuse Prompt with GPT-4 Completion**

```
I am playing with a set of blocks where I need to arrange the blocks into stacks. Here are the
↪  actions I can do

Pick up a block
Unstack a block from on top of another block
Put down a block
Stack a block on top of another block

I have the following restrictions on my actions:
I can only pick up or unstack one block at a time.
I can only pick up or unstack a block if my hand is empty.
I can only pick up a block if the block is on the table and the block is clear. A block is clear
↪  if the block has no other blocks on top of it and if the block is not picked up.
I can only unstack a block from on top of another block if the block I am unstacking was really on
↪  top of the other block.
I can only unstack a block from on top of another block if the block I am unstacking is clear.
Once I pick up or unstack a block, I am holding the block.
I can only put down a block that I am holding.
I can only stack a block on top of another block if I am holding the block being stacked.
I can only stack a block on top of another block if the block onto which I am stacking the block
↪  is clear.
Once I put down or stack a block, my hand becomes empty.
Once you stack a block on top of a second block, the second block is no longer clear.

[STATEMENT]
As initial conditions I have that, the red block is clear, the yellow block is clear, the hand is
↪  empty, the red block is on top of the blue block, the yellow block is on top of the orange
↪  block, the blue block is on the table and the orange block is on the table.
My goal is to have that the orange block is on top of the red block.
My plan is as follows:

[PLAN]
unstack the yellow block from on top of the orange block
put down the yellow block
pick up the orange block
stack the orange block on top of the red block
[PLAN END]

[STATEMENT]
As initial conditions I have that, the red block is clear, the yellow block is clear, the hand is
↪  empty, the red block is on top of the blue block, the yellow block is on top of the orange
↪  block, the blue block is on the table and the orange block is on the table.
My goal is to have that the red block is clear, the orange block is clear, the yellow block is
↪  clear, the hand is empty, the red block is on top of the blue block, the blue block is on the
↪  table, the orange block is on the table and the yellow block is on the table.
My plan is as follows:

[PLAN]
------------GPT-4 response----------------
unstack the yellow block from on top of the orange block
put down the yellow block
unstack the red block from on top of the blue block
put down the red block
[PLAN END]
------------Extracted LLM plan----------------
(unstack yellow orange)
(put-down yellow)
(unstack red blue)
(put-down red)

------------Ground truth plan----------------
(unstack yellow orange)
(put-down yellow)

====================SUCCESS====================
```

---

## A.11.2 Logistics

```
I have to plan logistics to transport packages within cities via trucks and between cities via
↪  airplanes. Locations within a city are directly connected (trucks can move between any two
↪  such locations), and so are the cities. In each city there is exactly one truck and each city
↪  has one location that serves as an airport.
Here are the actions that can be performed:

Load a package into a truck. For example, load package_1 into truck_1 at location_1_1.
Load a package into an airplane. For example, load package_1 into airplane_1 at location_1_1.
Unload a package from a truck. For example, unload package_1 from truck_1 at location_1_1.
Unload a package from an airplane. For example, unload package_1 from airplane_1 at location_1_1.
Drive a truck from one location to another location. For example, drive truck_1 from location_1_1
↪  to location_1_2 in city_1.
Fly an airplane from one city to another city. For example, fly airplane_1 from location_1_1 to
↪  location_2_1. Here location_1_1 is the airport in city_1 and location_2_1 is the airport in
↪  city_2.

The following are the restrictions on the actions:
A package can be loaded into a truck only if the package and the truck are in the same location.
Once a package is loaded into a truck, the package is not at the location and is in the truck.
A package can be loaded into an airplane only if the package and the airplane are in the same
↪  location.
Once a package is loaded into an airplane, the package is not at the location and is in the
↪  airplane.
A package can be unloaded from a truck only if the package is in the truck.
Once a package is unloaded from a truck, the package is not in the truck and is at the location of
↪  the truck.
A package can be unloaded from an airplane only if the package in the airplane.
Once a package is unloaded from an airplane, the package is not in the airplane and is at the
↪  location of the airplane.
A truck can be driven from one location to another if the truck is at the from-location and both
↪  from-location and to-location are locations in the same city.
Once a truck is driven from one location to another, it is not at the from-location and is at the
↪  to-location.
An airplane can be flown from one city to another if the from-location and the to-location are
↪  airports and the airplane is at the from-location.
Once an airplane is flown from one city to another the airplane is not at the from-location and is
↪  at the to-location.

[STATEMENT]
As initial conditions I have that, location_0_0 is an airport, location_1_0 is an airport,
↪  airplane_0 is at location_1_0, airplane_1 is at location_1_0, package_0 is at location_1_0,
↪  truck_0 is at location_0_0, truck_1 is at location_1_0, location_0_0 is in the city city_0 and
↪  location_1_0 is in the city city_1.
My goal is to have that package_0 is at location_0_0.
My plan is as follows:

[PLAN]
load package_0 into airplane_1 at location_1_0
fly airplane_1 from location_1_0 to location_0_0
unload package_0 from airplane_1 at location_0_0
[PLAN END]

[STATEMENT]
As initial conditions I have that, location_0_0 is an airport, location_1_0 is an airport,
↪  airplane_0 is at location_1_0, airplane_1 is at location_1_0, package_0 is at location_1_0,
↪  truck_0 is at location_0_0, truck_1 is at location_1_0, location_0_0 is in the city city_0 and
↪  location_1_0 is in the city city_1.
My goal is to have that location_0_0 is an airport, location_1_0 is an airport, airplane_0 is at
↪  location_1_0, airplane_1 is at location_0_0, truck_0 is at location_0_0, truck_1 is at
↪  location_1_0, location_0_0 is in the city city_0, location_1_0 is in the city city_1 and
↪  package_0 is in airplane_1.
My plan is as follows:

[PLAN]
------------Ground truth plan----------------
(load-airplane p0 a1 l1-0)
(fly-airplane a1 l1-0 l0-0)
```

### A.11.3 Mystery Blocksworld

---

**Mystery Blocksworld Plan Reuse Prompt with GPT-4 Completion**

```
I am playing with a set of objects. Here are the actions I can do

    Attack object
    Feast object from another object
    Succumb object
    Overcome object from another object

I have the following restrictions on my actions:
    To perform Attack action, the following facts need to be true: Province object, Planet object,
    ↪  Harmony.
    Once Attack action is performed the following facts will be true: Pain object.
    Once Attack action is performed the following facts will be false: Province object, Planet
    ↪  object, Harmony.
    To perform Succumb action, the following facts need to be true: Pain object.
    Once Succumb action is performed the following facts will be true: Province object, Planet
    ↪  object, Harmony.
    Once Succumb action is performed the following facts will be false: Pain object.
    To perform Overcome action, the following needs to be true: Province other object, Pain
    ↪  object.
    Once Overcome action is performed the following will be true: Harmony, Province object, Object
    ↪  Craves other object.
    Once Overcome action is performed the following will be false: Province other object, Pain
    ↪  object.
    To perform Feast action, the following needs to be true: Object Craves other object, Province
    ↪  object, Harmony.
    Once Feast action is performed the following will be true: Pain object, Province other object.
    Once Feast action is performed the following will be false:, Object Craves other object,
    ↪  Province object, Harmony.

[STATEMENT]
As initial conditions I have that, object a craves object b, object d craves object c, harmony,
↪  planet object b, planet object c, province object a and province object d.
My goal is to have that object c craves object a.
My plan is as follows:

[PLAN]
feast object d from object c
succumb object d
attack object c
overcome object c from object a
[PLAN END]

[STATEMENT]
As initial conditions I have that, object a craves object b, object d craves object c, harmony,
↪  planet object b, planet object c, province object a and province object d.
My goal is to have that object a craves object b, harmony, planet object b, planet object c,
↪  planet object d, province object a, province object c and province object d.
My plan is as follows:

[PLAN]
------------Ground truth plan----------------
(feast object d object c)
(succumb object d)
```

---

## A.12 Robustness to Goal Reformulation (Shuffling goal predicates) Prompts

### A.12.1 Blocksworld

---

**Blocksworld Robustness to Goal Reformulation (Shuffling goal predicates) Prompt with GPT-4 Completion**

```
I am playing with a set of blocks where I need to arrange the blocks into stacks. Here are the
↪  actions I can do

Pick up a block
Unstack a block from on top of another block
Put down a block
Stack a block on top of another block

I have the following restrictions on my actions:
```

---

```
I can only pick up or unstack one block at a time.
I can only pick up or unstack a block if my hand is empty.
I can only pick up a block if the block is on the table and the block is clear. A block is clear
↪  if the block has no other blocks on top of it and if the block is not picked up.
I can only unstack a block from on top of another block if the block I am unstacking was really on
↪  top of the other block.
I can only unstack a block from on top of another block if the block I am unstacking is clear.
Once I pick up or unstack a block, I am holding the block.
I can only put down a block that I am holding.
I can only stack a block on top of another block if I am holding the block being stacked.
I can only stack a block on top of another block if the block onto which I am stacking the block
↪  is clear.
Once I put down or stack a block, my hand becomes empty.
Once you stack a block on top of a second block, the second block is no longer clear.

[STATEMENT]
As initial conditions I have that, the red block is clear, the yellow block is clear, the hand is
↪  empty, the red block is on top of the blue block, the yellow block is on top of the orange
↪  block, the blue block is on the table and the orange block is on the table.
My goal is to have that the orange block is on top of the red block.

My plan is as follows:

[PLAN]
unstack the yellow block from on top of the orange block
put down the yellow block
pick up the orange block
stack the orange block on top of the red block
[PLAN END]

[STATEMENT]
As initial conditions I have that, the red block is clear, the yellow block is clear, the hand is
↪  empty, the red block is on top of the blue block, the yellow block is on top of the orange
↪  block, the blue block is on the table and the orange block is on the table.
My goal is to have that the orange block is on top of the red block.

My plan is as follows:

[PLAN]
------------GPT-4 response----------------
1. unstack the red block from on top of the blue block
2. put down the red block
3. unstack the yellow block from on top of the orange block
4. put down the yellow block
5. pick up the orange block
6. stack the orange block on top of the red block
[PLAN END]
------------Extracted LLM plan----------------
(unstack red blue)
(put-down red)
(unstack yellow orange)
(put-down yellow)
(pick-up orange)
(stack orange red)

------------Ground truth plan----------------

unstack the yellow block from on top of the orange block
put down the yellow block
pick up the orange block
stack the orange block on top of the red block
[PLAN END]

===================SUCCESS===================
```

## A.12.2   Logistics

```
I have to plan logistics to transport packages within cities via trucks and between cities via
↪  airplanes. Locations within a city are directly connected (trucks can move between any two
↪  such locations), and so are the cities. In each city there is exactly one truck and each city
↪  has one location that serves as an airport.
Here are the actions that can be performed:
```

```
Load a package into a truck. For example, load package_1 into truck_1 at location_1_1.
Load a package into an airplane. For example, load package_1 into airplane_1 at location_1_1.
Unload a package from a truck. For example, unload package_1 from truck_1 at location_1_1.
Unload a package from an airplane. For example, unload package_1 from airplane_1 at location_1_1.
Drive a truck from one location to another location. For example, drive truck_1 from location_1_1
↪  to location_1_2 in city_1.
Fly an airplane from one city to another city. For example, fly airplane_1 from location_1_1 to
↪  location_2_1. Here location_1_1 is the airport in city_1 and location_2_1 is the airport in
↪  city_2.

The following are the restrictions on the actions:
A package can be loaded into a truck only if the package and the truck are in the same location.
Once a package is loaded into a truck, the package is not at the location and is in the truck.
A package can be loaded into an airplane only if the package and the airplane are in the same
↪  location.
Once a package is loaded into an airplane, the package is not at the location and is in the
↪  airplane.
A package can be unloaded from a truck only if the package is in the truck.
Once a package is unloaded from a truck, the package is not in the truck and is at the location of
↪  the truck.
A package can be unloaded from an airplane only if the package in the airplane.
Once a package is unloaded from an airplane, the package is not in the airplane and is at the
↪  location of the airplane.
A truck can be driven from one location to another if the truck is at the from-location and both
↪  from-location and to-location are locations in the same city.
Once a truck is driven from one location to another, it is not at the from-location and is at the
↪  to-location.
An airplane can be flown from one city to another if the from-location and the to-location are
↪  airports and the airplane is at the from-location.
Once an airplane is flown from one city to another the airplane is not at the from-location and is
↪  at the to-location.

[STATEMENT]
As initial conditions I have that, location_0_0 is an airport, location_1_0 is an airport,
↪  airplane_0 is at location_1_0, airplane_1 is at location_1_0, package_0 is at location_1_0,
↪  truck_0 is at location_0_0, truck_1 is at location_1_0, location_0_0 is in the city city_0 and
↪  location_1_0 is in the city city_1.
My goal is to have that package_0 is at location_0_0.

My plan is as follows:

[PLAN]
load package_0 into airplane_1 at location_1_0
fly airplane_1 from location_1_0 to location_0_0
unload package_0 from airplane_1 at location_0_0
[PLAN END]

[STATEMENT]
As initial conditions I have that, location_0_0 is an airport, location_1_0 is an airport,
↪  airplane_0 is at location_1_0, airplane_1 is at location_1_0, package_0 is at location_1_0,
↪  truck_0 is at location_0_0, truck_1 is at location_1_0, location_0_0 is in the city city_0 and
↪  location_1_0 is in the city city_1.
My goal is to have that package_0 is at location_0_0.

My plan is as follows:

[PLAN]
------------Ground truth plan----------------

load package_0 into airplane_1 at location_1_0
fly airplane_1 from location_1_0 to location_0_0
unload package_0 from airplane_1 at location_0_0
[PLAN END]
```

### A.12.3   Mystery Blocksworld

Mystery Blocksworld Robustness to Goal Reformulation (Shuffling goal predicates) Prompt

```
I am playing with a set of objects. Here are the actions I can do

    Attack object
    Feast object from another object
    Succumb object
    Overcome object from another object
```

```
I have the following restrictions on my actions:
    To perform Attack action, the following facts need to be true: Province object, Planet object,
    ↪  Harmony.
    Once Attack action is performed the following facts will be true: Pain object.
    Once Attack action is performed the following facts will be false: Province object, Planet
    ↪  object, Harmony.
    To perform Succumb action, the following facts need to be true: Pain object.
    Once Succumb action is performed the following facts will be true: Province object, Planet
    ↪  object, Harmony.
    Once Succumb action is performed the following facts will be false: Pain object.
    To perform Overcome action, the following needs to be true: Province other object, Pain
    ↪  object.
    Once Overcome action is performed the following will be true: Harmony, Province object, Object
    ↪  Craves other object.
    Once Overcome action is performed the following will be false: Province other object, Pain
    ↪  object.
    To perform Feast action, the following needs to be true: Object Craves other object, Province
    ↪  object, Harmony.
    Once Feast action is performed the following will be true: Pain object, Province other object.
    Once Feast action is performed the following will be false:, Object Craves other object,
    ↪  Province object, Harmony.

[STATEMENT]
As initial conditions I have that, object a craves object b, object d craves object c, harmony,
↪  planet object b, planet object c, province object a and province object d.
My goal is to have that object c craves object a.

My plan is as follows:

[PLAN]
feast object d from object c
succumb object d
attack object c
overcome object c from object a
[PLAN END]

[STATEMENT]
As initial conditions I have that, object a craves object b, object d craves object c, harmony,
↪  planet object b, planet object c, province object a and province object d.
My goal is to have that object c craves object a.

My plan is as follows:

[PLAN]
------------Ground truth plan----------------

feast object d from object c
succumb object d
attack object c
overcome object c from object a
[PLAN END]
```

## A.13 Robustness to Goal Reformulation (Full → Partial) Prompts

### A.13.1 Blocksworld

Blocksworld Robustness to Goal Reformulation (Full → Partial) Prompt with GPT-4 Completion

```
I am playing with a set of blocks where I need to arrange the blocks into stacks. Here are the
↪  actions I can do

Pick up a block
Unstack a block from on top of another block
Put down a block
Stack a block on top of another block

I have the following restrictions on my actions:
I can only pick up or unstack one block at a time.
I can only pick up or unstack a block if my hand is empty.
I can only pick up a block if the block is on the table and the block is clear. A block is clear
↪  if the block has no other blocks on top of it and if the block is not picked up.
I can only unstack a block from on top of another block if the block I am unstacking was really on
↪  top of the other block.
I can only unstack a block from on top of another block if the block I am unstacking is clear.
Once I pick up or unstack a block, I am holding the block.
```

```
I can only put down a block that I am holding.
I can only stack a block on top of another block if I am holding the block being stacked.
I can only stack a block on top of another block if the block onto which I am stacking the block
↪  is clear.
Once I put down or stack a block, my hand becomes empty.
Once you stack a block on top of a second block, the second block is no longer clear.

[STATEMENT]
As initial conditions I have that, the red block is clear, the yellow block is clear, the hand is
↪  empty, the red block is on top of the blue block, the yellow block is on top of the orange
↪  block, the blue block is on the table and the orange block is on the table.
My goal is to have that the orange block is clear, the yellow block is clear, the hand is empty,
↪  the red block is on top of the blue block, the orange block is on top of the red block, the
↪  blue block is on the table and the yellow block is on the table.

My plan is as follows:

[PLAN]
unstack the yellow block from on top of the orange block
put down the yellow block
pick up the orange block
stack the orange block on top of the red block
[PLAN END]

[STATEMENT]
As initial conditions I have that, the red block is clear, the yellow block is clear, the hand is
↪  empty, the red block is on top of the blue block, the yellow block is on top of the orange
↪  block, the blue block is on the table and the orange block is on the table.
My goal is to have that the orange block is on top of the red block.

My plan is as follows:

[PLAN]
------------GPT-4 response----------------
unstack the yellow block from on top of the orange block
put down the yellow block
unstack the red block from on top of the blue block
put down the red block
pick up the orange block
stack the orange block on top of the red block
[PLAN END]
------------Extracted LLM plan----------------
(unstack yellow orange)
(put-down yellow)
(unstack red blue)
(put-down red)
(pick-up orange)
(stack orange red)

------------Ground truth plan----------------

unstack the yellow block from on top of the orange block
put down the yellow block
pick up the orange block
stack the orange block on top of the red block
[PLAN END]

===================SUCCESS====================
```

## A.13.2  Logistics

```
I have to plan logistics to transport packages within cities via trucks and between cities via
↪  airplanes. Locations within a city are directly connected (trucks can move between any two
↪  such locations), and so are the cities. In each city there is exactly one truck and each city
↪  has one location that serves as an airport.
Here are the actions that can be performed:

Load a package into a truck. For example, load package_1 into truck_1 at location_1_1.
Load a package into an airplane. For example, load package_1 into airplane_1 at location_1_1.
Unload a package from a truck. For example, unload package_1 from truck_1 at location_1_1.
Unload a package from an airplane. For example, unload package_1 from airplane_1 at location_1_1.
Drive a truck from one location to another location. For example, drive truck_1 from location_1_1
↪  to location_1_2 in city_1.
```

```
Fly an airplane from one city to another city. For example, fly airplane_1 from location_1_1 to
↪   location_2_1. Here location_1_1 is the airport in city_1 and location_2_1 is the airport in
↪   city_2.

The following are the restrictions on the actions:
A package can be loaded into a truck only if the package and the truck are in the same location.
Once a package is loaded into a truck, the package is not at the location and is in the truck.
A package can be loaded into an airplane only if the package and the airplane are in the same
↪   location.
Once a package is loaded into an airplane, the package is not at the location and is in the
↪   airplane.
A package can be unloaded from a truck only if the package is in the truck.
Once a package is unloaded from a truck, the package is not in the truck and is at the location of
↪   the truck.
A package can be unloaded from an airplane only if the package in the airplane.
Once a package is unloaded from an airplane, the package is not in the airplane and is at the
↪   location of the airplane.
A truck can be driven from one location to another if the truck is at the from-location and both
↪   from-location and to-location are locations in the same city.
Once a truck is driven from one location to another, it is not at the from-location and is at the
↪   to-location.
An airplane can be flown from one city to another if the from-location and the to-location are
↪   airports and the airplane is at the from-location.
Once an airplane is flown from one city to another the airplane is not at the from-location and is
↪   at the to-location.

[STATEMENT]
As initial conditions I have that, location_0_0 is an airport, location_1_0 is an airport,
↪   airplane_0 is at location_1_0, airplane_1 is at location_1_0, package_0 is at location_1_0,
↪   truck_0 is at location_0_0, truck_1 is at location_1_0, location_0_0 is in the city city_0 and
↪   location_1_0 is in the city city_1.
My goal is to have that location_0_0 is an airport, location_1_0 is an airport, airplane_0 is at
↪   location_1_0, airplane_1 is at location_0_0, package_0 is at location_0_0, truck_0 is at
↪   location_0_0, truck_1 is at location_1_0, location_0_0 is in the city city_0 and location_1_0
↪   is in the city city_1.

My plan is as follows:

[PLAN]
load package_0 into airplane_1 at location_1_0
fly airplane_1 from location_1_0 to location_0_0
unload package_0 from airplane_1 at location_0_0
[PLAN END]

[STATEMENT]
As initial conditions I have that, location_0_0 is an airport, location_1_0 is an airport,
↪   airplane_0 is at location_1_0, airplane_1 is at location_1_0, package_0 is at location_1_0,
↪   truck_0 is at location_0_0, truck_1 is at location_1_0, location_0_0 is in the city city_0 and
↪   location_1_0 is in the city city_1.
My goal is to have that package_0 is at location_0_0.

My plan is as follows:

[PLAN]
------------Ground truth plan----------------

load package_0 into airplane_1 at location_1_0
fly airplane_1 from location_1_0 to location_0_0
unload package_0 from airplane_1 at location_0_0
[PLAN END]
```

### A.13.3 Mystery Blocksworld

Mystery Blocksworld Robustness to Goal Reformulation (Full → Partial) Prompt

```
I am playing with a set of objects. Here are the actions I can do

    Attack object
    Feast object from another object
    Succumb object
    Overcome object from another object

I have the following restrictions on my actions:
    To perform Attack action, the following facts need to be true: Province object, Planet object,
    ↪   Harmony.
```

```
      Once Attack action is performed the following facts will be true: Pain object.
      Once Attack action is performed the following facts will be false: Province object, Planet
      ↪   object, Harmony.
      To perform Succumb action, the following facts need to be true: Pain object.
      Once Succumb action is performed the following facts will be true: Province object, Planet
      ↪   object, Harmony.
      Once Succumb action is performed the following facts will be false: Pain object.
      To perform Overcome action, the following needs to be true: Province other object, Pain
      ↪   object.
      Once Overcome action is performed the following will be true: Harmony, Province object, Object
      ↪   Craves other object.
      Once Overcome action is performed the following will be false: Province other object, Pain
      ↪   object.
      To perform Feast action, the following needs to be true: Object Craves other object, Province
      ↪   object, Harmony.
      Once Feast action is performed the following will be true: Pain object, Province other object.
      Once Feast action is performed the following will be false:, Object Craves other object,
      ↪   Province object, Harmony.

[STATEMENT]
As initial conditions I have that, object a craves object b, object d craves object c, harmony,
↪   planet object b, planet object c, province object a and province object d.
My goal is to have that object a craves object b, object c craves object a, harmony, planet object
↪   b, planet object d, province object c and province object d.

My plan is as follows:

[PLAN]
feast object d from object c
succumb object d
attack object c
overcome object c from object a
[PLAN END]

[STATEMENT]
As initial conditions I have that, object a craves object b, object d craves object c, harmony,
↪   planet object b, planet object c, province object a and province object d.
My goal is to have that object c craves object a.

My plan is as follows:

[PLAN]
------------Ground truth plan----------------

feast object d from object c
succumb object d
attack object c
overcome object c from object a
[PLAN END]
```

## A.14 Robustness to Goal Reformulation (Partial → Full) Prompts

### A.14.1 Blocksworld

---
**Blocksworld Robustness to Goal Reformulation (Partial → Full) Prompt with GPT-4 Completion**

```
I am playing with a set of blocks where I need to arrange the blocks into stacks. Here are the
↪   actions I can do

Pick up a block
Unstack a block from on top of another block
Put down a block
Stack a block on top of another block

I have the following restrictions on my actions:
I can only pick up or unstack one block at a time.
I can only pick up or unstack a block if my hand is empty.
I can only pick up a block if the block is on the table and the block is clear. A block is clear
↪   if the block has no other blocks on top of it and if the block is not picked up.
I can only unstack a block from on top of another block if the block I am unstacking was really on
↪   top of the other block.
I can only unstack a block from on top of another block if the block I am unstacking is clear.
Once I pick up or unstack a block, I am holding the block.
I can only put down a block that I am holding.
I can only stack a block on top of another block if I am holding the block being stacked.
```
---

```
I can only stack a block on top of another block if the block onto which I am stacking the block
↪  is clear.
Once I put down or stack a block, my hand becomes empty.
Once you stack a block on top of a second block, the second block is no longer clear.

[STATEMENT]
As initial conditions I have that, the red block is clear, the yellow block is clear, the hand is
↪  empty, the red block is on top of the blue block, the yellow block is on top of the orange
↪  block, the blue block is on the table and the orange block is on the table.
My goal is to have that the orange block is on top of the red block.

My plan is as follows:

[PLAN]
unstack the yellow block from on top of the orange block
put down the yellow block
pick up the orange block
stack the orange block on top of the red block
[PLAN END]

[STATEMENT]
As initial conditions I have that, the red block is clear, the yellow block is clear, the hand is
↪  empty, the red block is on top of the blue block, the yellow block is on top of the orange
↪  block, the blue block is on the table and the orange block is on the table.
My goal is to have that the orange block is clear, the yellow block is clear, the hand is empty,
↪  the red block is on top of the blue block, the orange block is on top of the red block, the
↪  blue block is on the table and the yellow block is on the table.

My plan is as follows:

[PLAN]
------------GPT-4 response----------------
unstack the yellow block from on top of the orange block
put down the yellow block
unstack the red block from on top of the blue block
put down the red block
pick up the orange block
stack the orange block on top of the red block
[PLAN END]
------------Extracted LLM plan----------------
(unstack yellow orange)
(put-down yellow)
(unstack red blue)
(put-down red)
(pick-up orange)
(stack orange red)

------------Ground truth plan----------------

unstack the yellow block from on top of the orange block
put down the yellow block
pick up the orange block
stack the orange block on top of the red block
[PLAN END]

===================SUCCESS====================
```

## A.14.2 Logistics

```
I have to plan logistics to transport packages within cities via trucks and between cities via
↪  airplanes. Locations within a city are directly connected (trucks can move between any two
↪  such locations), and so are the cities. In each city there is exactly one truck and each city
↪  has one location that serves as an airport.
Here are the actions that can be performed:

Load a package into a truck. For example, load package_1 into truck_1 at location_1_1.
Load a package into an airplane. For example, load package_1 into airplane_1 at location_1_1.
Unload a package from a truck. For example, unload package_1 from truck_1 at location_1_1.
Unload a package from an airplane. For example, unload package_1 from airplane_1 at location_1_1.
Drive a truck from one location to another location. For example, drive truck_1 from location_1_1
↪  to location_1_2 in city_1.
```

```
Fly an airplane from one city to another city. For example, fly airplane_1 from location_1_1 to
↪  location_2_1. Here location_1_1 is the airport in city_1 and location_2_1 is the airport in
↪  city_2.

The following are the restrictions on the actions:
A package can be loaded into a truck only if the package and the truck are in the same location.
Once a package is loaded into a truck, the package is not at the location and is in the truck.
A package can be loaded into an airplane only if the package and the airplane are in the same
↪  location.
Once a package is loaded into an airplane, the package is not at the location and is in the
↪  airplane.
A package can be unloaded from a truck only if the package is in the truck.
Once a package is unloaded from a truck, the package is not in the truck and is at the location of
↪  the truck.
A package can be unloaded from an airplane only if the package in the airplane.
Once a package is unloaded from an airplane, the package is not in the airplane and is at the
↪  location of the airplane.
A truck can be driven from one location to another if the truck is at the from-location and both
↪  from-location and to-location are locations in the same city.
Once a truck is driven from one location to another, it is not at the from-location and is at the
↪  to-location.
An airplane can be flown from one city to another if the from-location and the to-location are
↪  airports and the airplane is at the from-location.
Once an airplane is flown from one city to another the airplane is not at the from-location and is
↪  at the to-location.

[STATEMENT]
As initial conditions I have that, location_0_0 is an airport, location_1_0 is an airport,
↪  airplane_0 is at location_1_0, airplane_1 is at location_1_0, package_0 is at location_1_0,
↪  truck_0 is at location_0_0, truck_1 is at location_1_0, location_0_0 is in the city city_0 and
↪  location_1_0 is in the city city_1.
My goal is to have that package_0 is at location_0_0.

My plan is as follows:

[PLAN]
load package_0 into airplane_1 at location_1_0
fly airplane_1 from location_1_0 to location_0_0
unload package_0 from airplane_1 at location_0_0
[PLAN END]

[STATEMENT]
As initial conditions I have that, location_0_0 is an airport, location_1_0 is an airport,
↪  airplane_0 is at location_1_0, airplane_1 is at location_1_0, package_0 is at location_1_0,
↪  truck_0 is at location_0_0, truck_1 is at location_1_0, location_0_0 is in the city city_0 and
↪  location_1_0 is in the city city_1.
My goal is to have that location_0_0 is an airport, location_1_0 is an airport, airplane_0 is at
↪  location_1_0, airplane_1 is at location_0_0, package_0 is at location_0_0, truck_0 is at
↪  location_0_0, truck_1 is at location_1_0, location_0_0 is in the city city_0 and location_1_0
↪  is in the city city_1.

My plan is as follows:

[PLAN]
------------Ground truth plan----------------

load package_0 into airplane_1 at location_1_0
fly airplane_1 from location_1_0 to location_0_0
unload package_0 from airplane_1 at location_0_0
[PLAN END]
```

### A.14.3 Mystery Blocksworld

```
Mystery Blocksworld Robustness to Goal Reformulation (Partial → Full) Prompt

I am playing with a set of objects. Here are the actions I can do

    Attack object
    Feast object from another object
    Succumb object
    Overcome object from another object

I have the following restrictions on my actions:
    To perform Attack action, the following facts need to be true: Province object, Planet object,
    ↪  Harmony.
```

```
        Once Attack action is performed the following facts will be true: Pain object.
        Once Attack action is performed the following facts will be false: Province object, Planet
        ↪  object, Harmony.
        To perform Succumb action, the following facts need to be true: Pain object.
        Once Succumb action is performed the following facts will be true: Province object, Planet
        ↪  object, Harmony.
        Once Succumb action is performed the following facts will be false: Pain object.
        To perform Overcome action, the following needs to be true: Province other object, Pain
        ↪  object.
        Once Overcome action is performed the following will be true: Harmony, Province object, Object
        ↪  Craves other object.
        Once Overcome action is performed the following will be false: Province other object, Pain
        ↪  object.
        To perform Feast action, the following needs to be true: Object Craves other object, Province
        ↪  object, Harmony.
        Once Feast action is performed the following will be true: Pain object, Province other object.
        Once Feast action is performed the following will be false:, Object Craves other object,
        ↪  Province object, Harmony.

[STATEMENT]
As initial conditions I have that, object a craves object b, object d craves object c, harmony,
↪  planet object b, planet object c, province object a and province object d.
My goal is to have that object c craves object a.

My plan is as follows:

[PLAN]
feast object d from object c
succumb object d
attack object c
overcome object c from object a
[PLAN END]

[STATEMENT]
As initial conditions I have that, object a craves object b, object d craves object c, harmony,
↪  planet object b, planet object c, province object a and province object d.
My goal is to have that object a craves object b, object c craves object a, harmony, planet object
↪  b, planet object d, province object c and province object d.

My plan is as follows:

[PLAN]
------------Ground truth plan----------------

feast object d from object c
succumb object d
attack object c
overcome object c from object a
[PLAN END]
```