# OpenReview forum: "PlanBench: An Extensible Benchmark for Evaluating Large Language Models on Planning and Reasoning about Change"
_NeurIPS.cc/2023/Track/Datasets_and_Benchmarks — NeurIPS 2023 Datasets and Benchmarks Poster_

### Official Review · Reviewer_7obx · 2023-07-12
**A good LLM planning benchmark, but lacking in analysis**

**Rating:** 6
**Confidence:** 5
**Correctness:** The construction process is sound, an…
**Clarity:** Yes

**Strengths:**

It's very important for LLMs to have planning abilities; from this angle, this paper offers the following:

1. This paper offers a good taxonomy for looking at LLMs' planning abilities from different angles.
2. A very large number of prompts for each problem category.
3. Experiments results demonstrate the current performance for GPT4 and instructGPT models.

**Additional Feedback:**

N/A

**Documentation:**

N/A

**Opportunities For Improvement:**

1. The successful execution of LLMs highly depends on the prompt design; there needs to be more discussion on the alternative design choices of prompts. In order words, why do you use the proposed prompt? What components to include, and what is the impact of different pieces?

2. A comparison table highlighting the difference from other planning benchmarks would be nice.

3. In the related work section, the authors claim you are looking at the emergent planning abilities of LLMs. Therefore, they choose not to incorporate any environment feedback as prior works do. But, prior works have already shown environment feedback is extremely important to the success of LLM planners. Why do authors choose NOT to incorporate environment feedback? For humans, feedback loop is crucial for general decision-making. Even the classical planners have a sense of environment feedback e.g, cost functions, etc. This point is not necessarily a weakness, however, without feedback, the demonstrated result cannot reflect the planning abilities of LLMs to the full extent.

4. There needs to be more analysis of the results. As a benchmark paper, in addition to proposing a dataset. It's essential to provide some insights into why the proposed problem is complex. Instead of displaying numbers, showing LLMs are not working for certain categories of problems or working for specific problems. I would like to see some analysis. Like a summary of failure cases, some common attributes?

**Relation To Prior Work:**

Yes, but it could be better with a table.

**Summary And Contributions:**

This paper proposed a large-scale benchmark for LLM planning in a blocksworld and logistics domains. The paper offers a good taxonomy for looking at  LLMs' planning abilities from different angles and proposes corresponding tasks to evaluate LLMs planning abilities from these angles.  Then the paper shows comparisons of the results between GPT4 and instructGPT(davinci-002).

---

> ### Author Response · Authors · 2023-08-17
>
> We thank reviewer 7obx for their thoughtful comments. We are glad that the reviewer found our work to be a good taxonomy for LLMs planning abilities. We respond to the concerns of the reviewer below:
>
> > 1. The successful execution of LLMs highly depends on the prompt design; there needs to be more discussion on the alternative design choices of prompts. In order words, why do you use the proposed prompt? What components to include, and what is the impact of different pieces?
>
> Our prompt includes three major components, the domain description, example instances and the query instance. We believe that the domain description is important as the ability of being conditional on the prompt becomes crucial in cases where such general systems are being customized to specific domains. The example instances are provided for context (in terms of the distribution of the input text and the format) and then a query instance is presented. We are not restricting users of our benchmark from trying other prompting formats but we believe prompting configurations seem to have very little impact on the performance [1].
>
> > 2. A comparison table highlighting the difference from other planning benchmarks would be nice.
>
> To the best of our knowledge, we are the first to come up with a benchmark that evaluates the planning capabilities of Large Language Models. Previously, there were other benchmarks that evaluated other kinds of reasoning, but none were evaluating specifically planning or reasoning about actions and change.
>
> > 3. Why do authors choose NOT to incorporate environment feedback? For humans, feedback loop is crucial for general decision-making. Even the classical planners have a sense of environment feedback e.g, cost functions, etc. This point is not necessarily a weakness, however, without feedback, the demonstrated result cannot reflect the planning abilities of LLMs to the full extent.
>
> Our primary aim with this work was to put forward a benchmark for the community on the inherent planning capabilities of an LLM in an autonomous manner (similar to that of classical planners). Therefore, we chose to provide the same kind of information that we provide to classical planners (domain and problem information) in a human-readable format. We would like to distinguish two types of environmental feedback that the above comment seems to imply: environmental feedback reflecting new data (such as observations related to unexpected events or tasks failing) and environmental feedback further defining the domain (such as domain transition functions and cost functions). While we agree that we do not provide the former, PDDL domain and goal specification can be mapped to domain transition functions and a cost function. We do not argue that LLMs should be used only autonomously (see [1]) and our benchmark is extensible to the extent of including cases that take into account environment feedback as well. Further, we believe that it is critical to consider how the agent is allowed to use the feedback. In prior works, the agent conforms to an action once taken and receives feedback directly from the environment. This distinction is crucial in non-ergodic domains, as the agent may no longer be able to undo past actions when later feedback is received and receiving feedback from humans could be erroneous as humans themselves can be faulty verifiers.
>
> > 4. There needs to be more analysis of the results. As a benchmark paper, in addition to proposing a dataset. It's essential to provide some insights into why the proposed problem is complex. Instead of displaying numbers, showing LLMs are not working for certain categories of problems or working for specific problems. I would like to see some analysis. Like a summary of failure cases, some common attributes?
>
> We did do the analysis on obfuscation and the effect of variation in optimal plan length and # of objects. The obfuscation significantly worsens the performance of LLMs, and the plan length and # of objects don’t seem to be correlated with the LLM performance. We have added these results to the revised appendix (Appendix A.3 and A.4).
>
> Our working assumption was that benchmarks should focus on the characteristics of the benchmark, test suite and tasks, convince the reader that the benchmark is challenging for LLMs already, and encourage others to use the benchmark in evaluating their own research on the planning abilities of LLMs. We believe that our writeup does accomplish this.
>
> [1] S. Kambhampati, K. Valmeekam, M. Marquez & L. Guan. (2023, July). On the Role of Large Language Models in Planning. Tutorial presented at the International Conference on Automated Planning and Scheduling (ICAPS), Prague. https://yochan-lab.github.io/tutorial/ICAPS-2023/.

---

> > ### Comment · Reviewer_7obx · 2023-08-20
> >
> > "To the best of our knowledge, we are the first to come up with a benchmark that evaluates the planning capabilities of Large Language Models. Previously, there were other benchmarks that evaluated other kinds of reasoning, but none were evaluating specifically planning or reasoning about actions and change."
> >
> > Well, there is abundant work that evaluates the planning ability of LLMs; some are not in the style of PDDL.
> >
> > For example,
> > ALFWorld: Aligning Text and Embodied Environments for Interactive Learning.
> > Webshop: Towards scalable real-world web interaction with grounded language agents
> > MineDojo: Building Open-Ended Embodied Agents with Internet-Scale Knowledge.
> > VirtualHome: Simulating Household Activities via Programs
> >
> > In the original ALFRED paper, the planning sequences are generated using PDDL.
> >
> > There are a lot of papers using LLM to solve the planning problems in the above-listed benchmarks as well.
> > For example:
> > Language models as zero-shot planners: Extracting actionable knowledge for embodied agents
> >
> > ReAct: Synergizing Reasoning and Acting in Language Models
> >
> > Describe, Explain, Plan and Select: Interactive Planning with Large Language Models Enables Open-World Multi-Task Agents

---

> > > ### Author Response · Authors · 2023-08-21
> > >
> > > We thank the reviewer for their detailed response. We didn’t mean to say that there is no work in the intersection of LLMs and planning, but only that we are not aware of any direct *benchmarks* that look at a wide variety of planning capabilities as PlanBench does. Indeed we are aware of the works that the reviewer mentions; the paper cites React (reference number 30). More to the point, in a recent 3-hour long tutorial, we gave (URL https://bit.ly/3NC6vqs ), we point out that planning involves both gathering planning knowledge and resolving interactions among subplans. We then point out that many approaches such as React focus on retrieving planning knowledge as against interaction resolution. In contrast, PlanBench focuses on the interaction resolution in a variety of tasks involving planning and reasoning about plans.
> > >
> > > We will expand our related work section to make this point clearer.

---

> > > > ### Comment · Reviewer_7obx · 2023-08-25
> > > >
> > > > Please do.
> > > >
> > > > I think a discussion will help readers understand the paper better and it will help position the paper in the right place.
> > > > I don't have other issues with the paper. Thanks for the response.

---

### Official Review · Reviewer_nh2u · 2023-07-16
**PlanBench**

**Rating:** 7
**Confidence:** 5
**Correctness:** See above for critique of the evaluat…
**Clarity:** The paper is well written.

**Strengths:**

S1.  This is clearly an important space, as the authors point out, the work has already garnered numerous stars on GitHub.
S2.  The breakdown into finer tasks such as reasoning about plan execution, plan reuse, plan generalization etc is good - it gives a clearer indicator of exactly what LLMs are trying to do.
S3.  The obfuscation techniques provided are crucial, because LLMs have most likely seen many BlocksWorld problems - we actually confirmed they have based on our own work - they provide 'plans' without access to the domain.
S4.  A systematic manipulation of prompts is useful and can provide a much better understanding of LLM capabilities than reliance on a small number of prompts that tends to be an issue in this line of work.

**Additional Feedback:**

I think that addressing the issue of generalizing prompt generation from PDDL outside of any specific domain is critical for work in this space.  If the authors can address that, and perhaps describe their evaluation for those domains in detail, this would make for a very strong paper.

**Documentation:**

Yes, most of this work is derived from public planning problems that have been part of benchmarks in the planning community.

**Limitations:**

The authors suggest that more domains should be added (see W1) but to me the bigger problem is W2 which is what limits generalization to other domains easily.

**Opportunities For Improvement:**

W1.  The number of domains covered is small - (logistics and blocks world), and both have likely been seen heavily by many LLMs.  The obfuscation approach is a useful one here, but what would make this even more compelling is creating completely new domains for the benchmark.  That would test the out  of distribution capabilities of the benchmark better.
W2.  A template driven approach to converting PDDL to natural language seems very limiting - it surely results in rather stilted language, and also does not allow generalization to new domains.
W3.  I was somewhat surprised to see a rather limited evaluation on the benchmark - results were provided for the different tasks on BlocksWorld but not Logistics, results were not provided for the effects of obfuscation versus the original problems and so on (were all the problems obfuscated?). Results were also not provided for a parametric variation of the length of plans or the number of objects considered - all nice characteristics of the benchmark.  Did these variables have no distinctive effects?

The issues raised here seem to have been addressed in additional work.  Thank you.

**Relation To Prior Work:**

Yes

**Summary And Contributions:**

This paper describes PlanBench, a benchmark to evaluate the capabilities of LLMs for classical AI planning tasks.  The specific domains that are targeted are BlocksWorld and Logistics, both being well known and well studied domains in the AI planning community.  Most of these domains exist already and have been used heavily in the IPC community - the main contribution here is to break down the planning tasks of plan generation and cost optimal planning into a set of subtasks such as plan verification, reasoning about plan execution, replanning, etc.  The key contribution is to convert the planning problems as already defined in PDDL into natural language, provide prompts for these as well as have a templated mechanism to process the language back into a formal plan that can be verified by a planner.

---

> ### Author Response · Authors · 2023-08-17
>
> We thank reviewer nh2u for their thoughtful comments. We are glad that the reviewer found our work to be important, comprehensive and well written. We respond to the concerns of the reviewer below:
>
> > 1. The number of domains covered is small - (logistics and blocks world), and both have likely been seen heavily by many LLMs. The obfuscation approach is a useful one here, but what would make this even more compelling is creating completely new domains for the benchmark.
>
> Although we are aware that the training data of the LLMs might have contained these domains, our results on the blocksworld domain (which is a simple common-sense planning domain) indicate that the performance is quite inadequate for the majority of the test cases. But we do believe that the current data can always be fodder for future LLMs, increasing the importance of diagonalization in the testing. This is why we provide methods for obfuscating the domains by consistently replacing each domain object and predicate with either arbitrarily chosen meaning bearing words, or random strings.
>
> Currently, in our benchmark, along with the 2 domains and the obfuscated variants, we have already added another IPC domain (depots) to the Github repo and are in the process of adding one more (barman). We expect to complete adding it to the repo by this week. Additionally, we have included instructions on adding new domains and testing new LLMs in the github repo.
>
> > 2. A template driven approach to converting PDDL to natural language seems very limiting - it surely results in rather stilted language, and also does not allow generalization to new domains.
>
> We start by noting that the issue of the prompt format–whether it is in English or in some formal language (such as PDDL) is not expected to be critical to LLMs (see [1]). Obviously, giving the prompts in PDDL directly would avoid the templating step. The reason we allowed for prompts in quasi-English was more for the benefit of humans who might be involved in the loop–specifically end users who might either want to impose additional constraints on the plan, or have to “evaluate the plan” themselves. These two tasks are, after all, more naturally done in natural language. It is to telegraph this message to the end users that we allowed for prompts in templated language. The fact that the templated language may be somewhat stilted, and is perhaps not the way the end users would have written down the problem themselves is immaterial–as long as they realize that they can read the plans and evaluate them or add constraints. We have revised the paper to make this motivation more explicit (lines 191-196).
>
> [1] S. Kambhampati, K. Valmeekam, M. Marquez & L. Guan. (2023, July). On the Role of Large Language Models in Planning. Tutorial presented at the International Conference on Automated Planning and Scheduling (ICAPS), Prague. https://yochan-lab.github.io/tutorial/ICAPS-2023/.
>
> > 3. I was somewhat surprised to see a rather limited evaluation on the benchmark - results were provided for the different tasks on BlocksWorld but not Logistics, results were not provided for the effects of obfuscation versus the original problems and so on (were all the problems obfuscated?). Results were also not provided for a parametric variation of the length of plans or the number of objects considered - all nice characteristics of the benchmark. Did these variables have no distinctive effects?
>
> We did do the analysis on obfuscation and the effect of variation in plan length and # of objects. The obfuscation significantly worsens the performance of LLMs, and the plan length and # of objects don’t seem to be correlated with the LLM performance. We have added these results to the revised appendix (Appendix A.3 and A.4).
>
> Our working assumption was that benchmarks should focus on the characteristics of the benchmark, test suite and tasks, convince the reader that the benchmark is challenging for LLMs already, and encourage others to use the benchmark in evaluating their own research on the planning abilities of LLMs. We believe that our writeup does accomplish this.

---

> > ### Comment · Reviewer_nh2u · 2023-08-18
> >
> > Thank you for your response, I especially liked the fact that you addressed the issues I raised.

---

### Official Review · Reviewer_YvuN · 2023-07-21
**Review of PlanBench**

**Rating:** 7
**Confidence:** 3

**Strengths:**

**Significance**
Planning capabilities appear to be a significant dimension of LLM reasoning capabilities.

**Relevance**
The paper appears relevant to the research community. It contributes meaningfully to a significant literature seeking to probe and understand LLMs. The interest this benchmark was able to generate in the past is an indication for its relevance.

**Quality**
The paper clearly lays out a framework for the (for most readers potentially unfamiliar) topic of planning evaluation. The authors convincingly lay out how they address a gap in the existing literature on LLM evaluation.

**Ethical and Social Implications**
I don't see relevant ethical or social implications that would pose a challenge to the authors and their contributions.


**Additional Feedback:**

-

**Clarity:**

Generally, the paper is very well written. At the same time, I recommend dedicating at least a few lines of the main paper to a concrete example of planning tasks. From the main text alone, without reading the appendix, the general level of abstraction in the paper can make it difficult for readers outside the specific subfield to develop an intuitive understanding of the authors' definition of a planning task. The paper is of broad relevance, but my concern is that this broad relevance may not be apparent to most potential readers.

**Correctness:**

The two models in this paper (GPT4 and Instruct-GPT3) are generally evaluated in a few-shot learning setting in which the models are provided with a few examples. Maybe I have missed it, but it seems to me as if the authors used the same examples throughout, inducing a potential source of unaccounted-for variation in the evaluation process. Consistent with this assumption, they state that the reporting of error bars is not appropriate for their study (Checklist 3(c)).

While I understand that it would be too costly to evaluate across all possible combinations of training and evaluation sets, I do think it would be instructive to at least give readers a sense of the sensitivity of the performance to the training examples, e.g. on a small subsample of the tasks which are evaluated repeatedly using different examples. In suggesting this, I am aware that that evaluation is just a small part of the paper. At the same time, it would be good to set a positive example, should the benchmark be adopted more widely.

Two other choices that I feel need to at least be justified:

- In evaluating plan generations, the authors omit instances of missing plan-end tags or failed extractions. I would have assumed these should be counted as failed attempts at accurate plan generation.

- In evaluating plan verification, the authors only require the model to identify one missing precondition. That seems sensible, but still requires justification, if only in a footnote.

Last but not least, I think it might be worth at least briefly pointing out the substantial difference in the performance of GPT-4 and Instruct-GPT3.

**Documentation:**

The documentation is adequate.

**Ethics:**

I do not suspect ethical concerns with this submission.

**Limitations:**

See below

**Opportunities For Improvement:**

See below

**Relation To Prior Work:**

The authors explain how their paper is related to previous contributions.

That said, I believe the greatest weakness of the paper is in its (lack of) an explanation of its motivation, including for how its interest of study (planning) concretely contributes to solving more applied problems. On lines 38-40, they motivate their paper through a reference to planning as a "core aspect of human intelligence" and the relevance to applied problems in "automatic code generation, moral and even deontological reasoning". I find this motivation insufficiently precise, especially because the authors chose to include in their list of motivations two candidates (moral and deontological reasoning) that will likely surprise most readers. I urge the authors to be more specific and elaborate.

**Summary And Contributions:**

The paper introduces a new benchmark that can be used to evaluate language models on planning tasks. It familiarizes readers with a formal framework to think about reasoning tasks, and then varies individual components within that framework to create a large number of prompt posing planning challenges. The individual components correspond to specific aspects of the planning and plan-execution process, such as generating a plan or verifying whether a provided plan is executable and if not, what a deficiency is.

---

> ### Author Response · Authors · 2023-08-17
>
> We thank reviewer YvuN for their thoughtful comments. We are glad that the reviewer found our work to be relevant and significant. We respond to the concerns of the reviewer below:
>
> > 1. That said, I believe the greatest weakness of the paper is in its (lack of) an explanation of its motivation, including for how its interest of study (planning) concretely contributes to solving more applied problems.
>
> One of the most pressing questions about LLM capabilities currently is whether they are just doing a form of approximate retrieval, or whether they have emergent reasoning/planning capabilities. Most of the claims on either side of this debate have been anecdotal rather than systematic (see [1]). This benchmark is an attempt to provide a systematic way to empirically evaluate the planning abilities of LLMs in an automated fashion. As such, we believe that the benchmark is thus well motivated in the current zeitgeist. The use of this benchmark may also lead to the development of LLM extensions—the so-called LLM Modulo approaches–that may wind up marrying the advantages of LLMs (in generating guesses) and first principles reasoners (which provide guarantees of their verification abilities).
>
> [1] S. Kambhampati, K. Valmeekam, M. Marquez & L. Guan. (2023, July). On the Role of Large Language Models in Planning. Tutorial presented at the International Conference on Automated Planning and Scheduling (ICAPS), Prague. https://yochan-lab.github.io/tutorial/ICAPS-2023/.
>
> > 2. Maybe I have missed it, but it seems to me as if the authors used the same examples throughout, inducing a potential source of unaccounted-for variation in the evaluation process. Consistent with this assumption, they state that the reporting of error bars is not appropriate for their study (Checklist 3(c)). While I understand that it would be too costly to evaluate across all possible combinations of training and evaluation sets, I do think it would be instructive to at least give readers a sense of the sensitivity of the performance to the training examples, e.g. on a small subsample of the tasks which are evaluated repeatedly using different examples. In suggesting this, I am aware that that evaluation is just a small part of the paper. At the same time, it would be good to set a positive example, should the benchmark be adopted more widely.
>
> No, we do not give the same examples for each test problem. In fact, the correct plan for n-th test problem is given as the example for (n+1)-th test problem. Thus the reviewer’s concern doesn’t hold. While we could, of course, try running the same test case multiple times with different examples, our approach of giving an effectively random example for each test case, and looking at the performance on 600 test cases does give us the same analytical insights.
>
> Regarding the error bars, all we meant to say was that since we are running each test case only once, there was no variation to be reported.
>
> > 3. Two other choices that I feel need to at least be justified:
>     - In evaluating plan generations, the authors omit instances of missing plan-end tags or failed extractions. I would have assumed these should be counted as failed attempts at accurate plan generation.
>     - In evaluating plan verification, the authors only require the model to identify one missing precondition. That seems sensible, but still requires justification, if only in a footnote.
>
> We start by noting that we didn’t encounter a single case in our plan generation experiments with missing plan-end tags or failed extractions. So, the results are not actually affected. That said, the reason we had originally put a plan-end tag (as separate from end-generation tag) was to allow for cases where the LLM gives the plan, and also adds commentary on the plan (explanations etc). We introduced the plan-end tag to make the life of the plan extractor easy in such cases; the plan-end tag does not function as an additional format “quiz” for the LLM and is irrelevant if not needed by the extractor. As it happened, in most of our cases, the plan-end and end-generation tags happened to be the same. The tags themselves are therefore irrelevant in the eyes of the extractor and are ignored. In a rare (and currently unseen) case where the extractor runs into an error while extracting, we will not attempt to evaluate a plan and instead mark that instance as incorrect. We have revised the paper to reflect this (lines 143-148).
>
>
>
> The evaluation method for plan verification requires the model to identify at least one missing precondition from the first action in the plan that fails if the plan is inexecutable. This information is a sufficient condition to explain why a plan is invalid. Asking the LLM to come up with this minimum information would give us a better understanding of why the LLM might have marked the plan as invalid.

---

> > ### Comment · Reviewer_YvuN · 2023-08-29
> >
> > Thanks for your comments.
> >
> > > One of the most pressing questions about LLM capabilities currently is whether they are just doing a form of approximate retrieval, or whether they have emergent reasoning/planning capabilities. Most of the claims on either side of this debate have been anecdotal rather than systematic (see [1]). This benchmark is an attempt to provide a systematic way to empirically evaluate the planning abilities of LLMs in an automated fashion. As such, we believe that the benchmark is thus well motivated in the current zeitgeist. The use of this benchmark may also lead to the development of LLM extensions—the so-called LLM Modulo approaches–that may wind up marrying the advantages of LLMs (in generating guesses) and first principles reasoners (which provide guarantees of their verification abilities).
> >
> > I don't doubt that the benchmark can be well motivated, but my argument is that the paper doesn't make that case. In fact, I'd argue that, if it tries to provide a motivation, the paper focuses on some of the most contestable reasons to care about planning. The only sentence I see that clearly motivates a focus on planning (as opposed to other forms of reasoning) is on p.2:
> >
> > "Our focus on planning is spurred by not only the fact that it is a core aspect of human intelligence, but also that it is required for many of the tasks considered as potential applications of LLMs including automatic code generation, moral and even deontological reasoning [6]".
> >
> > The provided reference for this statement doesn't even talk about planning in any way.
> >
> > This is all fine if the authors see themselves in communication with domain-experts only, as those don't need further motivation to care. But it seems to me that there is a low-hanging fruit here to make the benchmark more broadly appealing without incurring any cost.
> >
> > > No, we do not give the same examples for each test problem. In fact, the correct plan for n-th test problem is given as the example for (n+1)-th test problem. Thus the reviewer’s concern doesn’t hold. While we could, of course, try running the same test case multiple times with different examples, our approach of giving an effectively random example for each test case, and looking at the performance on 600 test cases does give us the same analytical insights.
> >
> > Thanks for the clarification. I don't think what the paper does is comparable to what I suggested. Although I realize my description could have been more clear, I meant to suggest a type of bootstrapping procedure, which will give readers insight into the sampling error from a larger reference population (here the universe of all possible plans that fit the described process), and does, in fact, allow the authors to report uncertainty surrounding their estimates. I believe it is relevant to examine to what extent the results are dependent on the particular plans that were included in the analysis, because this will help assess the reliability of out-of-sample inference. The paper's process simply ensures that there is no systematic dependence between the example and the test case, but otherwise represents a single draw of examples and associated test cases. As such, it is not possible to quantify uncertainty, whether induced by the sampling process or the choice of examples and test cases. That's fine, this is a benchmark paper, not a model evaluation paper, but I do think readers interested in the evaluation miss out on important information.

---

### Official Review · Reviewer_vfR7 · 2023-07-21
**A new benchmark for evaluating the planning abilities of LLMs.**

**Rating:** 8
**Confidence:** 4

**Strengths:**

The benchmark’s motivation is solid and clearly explained. PlanBench is complex and involves many components, which are adequately covered in the text.

PlanBench covers a large number of different test cases, which researchers should find especially useful. Much of the paper is dedicated to explaining these test cases in detail, and presenting experimental results for each one.


**Additional Feedback:**

It would be useful to potential users if the repo listed the papers that have used PlanBench.

**Clarity:**

The writing is mostly clear and easy to follow. Many grammatical flaws remain (like “the benchmark could be extended to support *more number* of IPC domains.”) but these should be easy to correct.

**Correctness:**

The benchmark design and its test framework are comprehensive, and the eight separate test cases are well thought out. I see no methodological problems.

**Documentation:**

The repository is well organized and thorough. It provides installation requirements, and complete run commands with argument explanations for reproducing all major experiments in the paper. Support is provided for many current LLMs. The code for problem generation and obfuscation are also included. It’s especially useful that the repo explains how to add new LLMs and IPC domains to the benchmark.

**Ethics:**

No concerns.

**Limitations:**

No concerns.

**Opportunities For Improvement:**

The abstract highlights the benchmark’s “diversity”, but two domains doesn’t seem particularly diverse. It’s the bare minimum for a multi-domain benchmark, and a good start, but the full value of PlanBench will depend on more domains being added in the future.

**Relation To Prior Work:**

Related existing benchmarks are listed, and their limitations are clearly explained.

**Summary And Contributions:**

This paper presents PlanBench to support evaluation of the planning and reasoning abilities of LLMs. The benchmark currently covers two IPC domains (Blocksworld and Logistics), and is designed to allow the addition of more domains. Two prominent LLMs (GPT-4 and text-davinci-002) are evaluated on hundreds of Blocksworld tasks, demonstrating that while GPT-4 significantly outperforms text-davinci-002, it still falls well short of fully mastering most tasks.

---

> ### Author Response · Authors · 2023-08-17
>
> We thank reviewer vfR7 for their valuable comments. We are glad that the reviewer found our work to be solid, comprehensive and well organized. We respond to the concerns of the reviewer below:
>
> > 1. The abstract highlights the benchmark’s “diversity”, but two domains doesn’t seem particularly diverse. It’s the bare minimum for a multi-domain benchmark, and a good start, but the full value of PlanBench will depend on more domains being added in the future.
>
> We would like to point out that along with the 2 domains and the family of additional domains that can be generated by obfuscation techniques, we have already added another IPC domain (depots) to the Github repo and are in the process of adding one more (barman) which we expect to complete by this week. Additionally, we have included instructions on adding new domains and testing new LLMs in the github repo.

---

> > ### Comment · Reviewer_vfR7 · 2023-08-26
> > **Response**
> >
> > Thanks for the comments. Great to see more domains being added!

---

### Decision · Program_Chairs · 2023-09-22

**Decision:**

Accept (Poster)

**Comment:**

The paper proposes a benchmark for evaluating the ability of LLMs to plan and reason on classical AI problems (BlocksWorld and Logistics).  Overall, all reviewers supports accepting the paper.  The paper is mostly well written (although R-vfR7 noted some grammar issues).  Reviewers noted that many LLMs may have seen the classical tasks and appreciated the effort the authors went to to provide "obsfucated" versions of the tasks.  The AC agrees there is value to this work and recommends acceptance.

The authors are encouraged to incorporate reviewers suggestions for their camera ready including the following:
1. Clarify explanation of motivation for the benchmark (YvuN)
2. Expand related work to clarify difference from this work to some other work that uses LLMs for plannings (7obx)
3. Add concrete examples of tasks to main paper (YvuN)
4. Proofreading and fixing various grammatical errors (L44: ';' => ':', L285,L291: missing period)
5. Update paper to mention that additional tasks/domains are actively being added.